

# Automated prioritizing heuristics for parallel task graph scheduling in heterogeneous computing

Clément Flint[1,2,3], Ludovic Paillat[1,2,3] and Bérenger Bramas[1,2,3]

[1] ICPS Team, ICube Laboratory, Illkirch Graffenstaden, Grand Est, France
[2] CAMUS Team, Inria Nancy, Nancy, Grand Est, France
[3] Department of Mathematics and Computer Science, University of Strasbourg, Strasbourg, Grand Est, France

## ABSTRACT

High-performance computing (HPC) relies increasingly on heterogeneous hardware and especially on the combination of central and graphical processing units. The task-based method has demonstrated promising potential for parallelizing applications on such computing nodes. With this approach, the scheduling strategy becomes a critical layer that describes where and when the ready-tasks should be executed among the processing units. In this study, we describe a heuristic-based approach that assigns priorities to each task type. We rely on a fitness score for each task/worker combination for generating priorities and use these for configuring the Heteroprio scheduler automatically within the StarPU runtime system. We evaluate our method's theoretical performance on emulated executions and its real-case performance on multiple different HPC applications. We show that our approach is usually equivalent or faster than expert-defined priorities.

## INTRODUCTION

Heterogeneous computing refers to the use of different kinds of processing units within a node. This type of hardware is widespread in the the high-performance computing (HPC) world and equips several of the fastest supercomputers, such as Summit which is ranked second in 2021 according to TOP500 (*Hans et al., 2021*). Among these, most heterogeneous systems are composed of central processing units (CPUs) and graphical processing units (GPUs). Developing efficient applications for this type of node is challenging because it requires managing the memory transfers and the load balancing between the processing units. Thus, the HPC community invested much effort into designing programming models to relieve the developers from managing these complex issues.

The task-based model has demonstrated high potential in various fields (*Agullo et al., 2014*, *2015a*; *Carpaye, Roman & Brenner, 2018*). In this method, an algorithm is divided into tasks and data accesses from which a directed acyclic graph (DAG) is deduced. The nodes in this DAG represent tasks, and the edges represent their dependencies. The runtime system is in charge of abstracting the machinery and ensures a coherent parallel execution of these DAGs. Two examples of runtime systems which handle heterogeneous

Corresponding author
Clément Flint, clement.flint@inria.fr

workloads are Parsec (*Bosilca et al., 2013*) and StarPU (*Augonnet et al., 2011*), In this paradigm, the scheduler decides which worker executes which ready-task. Heteroprio (*Agullo et al., 2016*) is a scheduler that is implemented in StarPU. It has been designed for heterogeneous machines and is used by several applications where it provides significant improvements (*Agullo et al., 2015b*; *Lopez & Duff, 2018*). However, users must tune Heteroprio by providing priorities for the different types of tasks that exist in their applications. Heteroprio, consequently, requires more programming effort from the user compared to most schedulers. It also relies on costly benchmarks or on correct programmer intuition about task priorities. In both cases, the choice of priorities can lead to inefficient scheduling decisions. Finally, the definition of static priorities prevents any dynamic adaptation throughout the execution.

In this study, we aim to create a method that automatically computes efficient priorities for Heteroprio. The main focus here lies in the automation of Heteroprio. Achieving high-performance is a secondary objective. We propose different heuristics that provide a fitness score for each combination task type/processing unit. These scores allow us to deduce the priorities by sorting each processing unit and the task types by descending score. The contributions of this study are as follows:

- we describe different heuristics that lead to efficient priorities;
- we define a new methodology for configuring the Heteroprio scheduler automatically according to these automatic priorities;
- we evaluate our approach on a wide range of graphs using emulated executions;
- we validate our concept in StarPU by running existing task-based scientific applications with our new automatic scheduler.

These contributions lead to a new version of Heteroprio in StarPU referred to as AutoHeteroprio. AutoHeteroprio can be considered as a fully automatic scheduler, whereas Heteroprio is semi-automatic. Moreover, we show that using the fully automatic version does not induce significant slowdowns and may sometimes lead to speedups.

The article is organized as follows. In "Background", the background and prerequisite are described. We define the problem of task scheduling, we present related works, we introduce Heteroprio and we formalize the problem we target in our study. In "Heuristics for automatic configuration", we present various heuristics and implementation details. Finally, in "Performance study", the evaluation of the performance of the approach is presented.

# BACKGROUND

## Scheduling problem

The objective of the task graph scheduling problem is usually to minimize the overall program finish time (*i.e.*, makespan). This finish time depends on the sequence in which the tasks are executed and on their affectation to a processor type (*Kwok & Ahmad, 1999*). There are variations of the finish time objective. For example, some research aims at reducing the mean finish time (MFT), also known as the mean time of a system or the

mean flow time, which is the average finish time of all of the tasks executed (*Bruno, Coffman & Sethi, 1974*; *Leung & Young, 1989*). The MFT criterion tends to minimize the memory required to hold the incomplete tasks. Some works aim at improving other metrics, such as energy consumption (*Zhou et al., 2016*). The overall finish time, however, remains the most often used metric in scheduling and this is why we use it for measuring performance.

### Related work

In the context of heterogeneous computing, it has been proven that finding an optimal schedule is NP-complete in the general case (*Brucker & Knust, 2009*). Therefore, the research community has proposed various schemes whose goals are to obtain efficient executions. There are two typical ways of performing scheduling: statically or dynamically. In static scheduling, the decisions are computed before the execution, whereas in dynamic scheduling, the scheduler takes decisions throughout the execution of the application. There can be different degrees of static and dynamic scheduling, and some studies describe hybrid static/dynamic strategies (*Donfack et al., 2011*).

*Yu-Kwong & Ahmad (1996)* presented a static scheduling which distributes the workload on fully connected multiprocessors. This algorithm is known as the dynamic critical-path scheduling algorithm and relies on the computation of a critical path. In practice, the high-performance community tends towards dynamic rather than static scheduling. One of the reasons for this is that some complex dependencies cannot be represented by a DAG. This algorithm can, therefore, not be used to its full potential in most modern applications. *Topcuoglu, Hariri & Min-You (2002)* presented the Heterogeneous Earliest-Finish-Time (HEFT) and the CPOP algorithms. HEFT relies on the computation of the earliest finish time (EFT). It prioritizes tasks that minimize the EFT. The critical path on processor algorithm differs from HEFT in the manner it computes the critical paths for each processor, which lets it estimate communication costs and takes them into account for its scheduling. HEFT has become a widespread algorithm and is implemented in most execution engines. The original implementation of this scheduler, however, needs to analyze the task graph in its entirety. This results in a significant overhead that increases as the graph grows in size. *Khan (2012)* introduced the constrained earliest finish time (CEFT) algorithm. This method adds the concept of constrained critical paths (CCPs), which are small task windows representing ready tasks at one instance. The CEFT algorithm usually performs better than the original HEFT algorithm but has the same major bottlenecks. *Jiang, Shao & Guo (2014)* explored the possibility of using Tuple-Based Chemical Reaction Optimization to perform scheduling. Their implementation typically produces results comparable to those of HEFT. *Choi et al. (2013)* proposed a dynamic scheduling that relies on a history-based Estimated-Execution-Time (EET) for each task. The idea of this algorithm is to schedule each task on its fastest architecture. In some cases, the scheduler ignores this rule and executes a task on a slower processor (*e.g.*, in the case of work starvation for a worker type). *Xu et al. (2014)* introduced an efficient genetic algorithm for heterogeneous scheduling. This algorithm achieves performance comparable to HEFT variants and CPOP on their test cases. The

drawback of current state-of-the-art genetic-based schedulers is that they typically require large-sized DAGs or may require multiple repeats to reach their full efficiency. *Wen, Wang & O'Boyle (2014)* computed the relative CPU and GPU speedups and directly use this metric to compute the priority of the tasks. This approach is notably efficient when data transfers are low. There are, however, cases where the speedup predictor is not sufficient for performing efficient scheduling. *Luo et al. (2021)* used a specific graph convolutional network to analyze the DAG and infer the probability of executing each task. Additionally, multiple agents are used for defining the scheduling strategy. A reinforcement learning algorithm improves these agents over the execution period. This method does produce efficient scheduling in simulators, but it is unlikely to be applicable in real-time executions. We refer to the works of *Maurya & Tripathi (2018)* and *Beaumont et al. (2020)* which provide surveys of several classical schedulers.

### Heteroprio overview

The Heteroprio scheduler has been developed for optimizing the fast multipole method and is implemented in StarPU (*Bramas, 2016*; *Agullo et al., 2016*; *Augonnet et al., 2011*). StarPU is designed such that the scheduler is a distinct component that a user can change or customize. StarPU schedulers rely on two mechanisms known as push-task and pop-task. The push-task is called when a task becomes ready (*i.e.*, when all its dependencies are satisfied). The workers indirectly provoke this call at the end of the execution of a task, if it does allow a new task to be executed. A worker calls the pop function when it fetches a task. This happens either because it has just finished executing a task or after it has been idle for a certain amount of time. Thus, in StarPU, the behavior of a scheduler can be summarized by its push and pop mechanisms.

Heteroprio uses multiple lists of buckets. Each bucket is a first in, first out (FIFO) queue of tasks. When a task becomes available, it is pushed to a bucket. The target bucket is set by the user when submitting the task. There is typically one bucket per task type but the user can choose to group the tasks as they wish. Besides, each architecture has a priority list that represents the order in which the corresponding workers access the buckets. When a worker becomes available, it iterates over the buckets using the priority list and picks a task from the first non-empty bucket it finds. Therefore, these lists define which tasks are favored by a particular architecture. The user must fill them before the beginning of the parallel execution. Figure 1 schematizes how the workers select their tasks in Heteroprio. For the sake of simplicity, the CPU and GPU priorities are mirrored, but this is not necessarily the case: we can apply any permutation to the priority list of a processor type.

We provide a detailed example of an execution with Heteroprio in the "Heteroprio execution example". In 2019, an enhancement has been brought to Heteroprio to take into account the data locality (*Bramas, 2019*). The original version treats all workers of the same type exactly equally, which completely discards memory management and can lead to massive and sometimes avoidable data movement. In the new version of Heteroprio, known as LaHeteroprio, workers select their tasks not only depending on their position in the FIFO list of the buckets but also depending on their memory affinity with the tasks. The affinity is computed thanks to multiple heuristics that the user can choose.

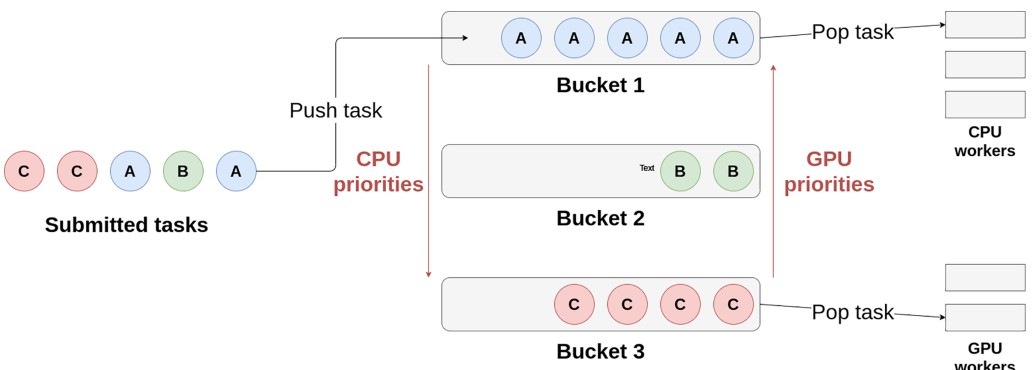

**Figure 1 Schema of the principle of Heteroprio.** The CPU workers iterate on buckets 1, then 2, and finally 3. The GPU workers iterate the other way around in this example.

## Formalization

### *General scheduling problem*

The scheduling problem is usually defined as follows. Let us consider an application that has a matching DAG referred to as $G = (V, E)$, where $V$ are the $v$ nodes and $E$ are the edges. Each node represents a task, and each edge represents a dependency between two tasks. We define $Q$ as the set of $q$ processors and $W$ as the computation cost matrix. The nodes (tasks) are referred to as $v_i$, where $i$ can range from 1 to $v$. The processors are referred to as $p_j$, where $j$ can range from 1 to $q$. This computation cost matrix is of size $v \times q$ and $w_{i,j}$ represents the cost of executing task $v_i$ on processor $p_j$. The cost can be any metric that we seek to minimize. In our case, it is the execution time of a task.

To take data transfers into account, we can add the following definitions. The *Data* matrix represents the required data transfers. $Data_{i,k}$ is the amount of data that needs to be transferred from the processor that executes $v_i$ to the processor that executes $v_k$. The $B$ matrix defines the transfer rates between processors: $B_{i,k}$ is the transfer rate between $p_i$ and $p_k$. The $L$ vector represents the communication startup cost of each processor. Hence, the model allows us to define the communication cost of one edge $(i, k)$:

$$c_{i,k} = L_m + \frac{Data_{i,k}}{B_{m,n}}, \tag{1}$$

where $m$ and $n$ represent, respectively, the chosen processor for $v_i$ and $v_k$.

To provide a formal definition of the makespan, we introduce the Actual Start Time, and the Actual Finish Time (AST, and AFT). The AFT of a task $v_i$ is defined by $AFT(v_i) = AST(v_i) + w_{i,j}$ (where $p_j$ is the chosen processor for task $v_i$). The AST of a task $v_i$ is defined as follows:

$$AST(v_i) = \max_{v_j \in pred(v_i)}(AFT(v_j) + c_{j,i}), \tag{2}$$

where $pred(v_i)$ is the set of predecessors of $v_i$. This formula expresses that the task $v_i$ starts as soon as possible, but after all the transfers have been completed. The memory transfers can be ignored by removing the $c_{j,i}$ term.

The *schedule length* (or *makespan*) is defined as the finish time of the last task:

$$makespan = \max_{v_i \in V}(AFT(v_i)). \tag{3}$$

We define this makespan as our objective and aim to minimize it. The formalization we provide in this section is general and applicable to most scheduling situations. In the next section, we define additional notations and constraints that relate to the use of Heteroprio.

### Heteroprio automatic configuration problem

In this section, we present additional definitions that are needed for the specific Heteroprio scheduling problem. We define a set of $b$ buckets referred to as $b_i$, where $i$ can range from 1 to $b$. The concept of bucket is explained in "Heteroprio overview". A solution is defined by a matrix $S$, where $S_{i,j}$ is the priority of task $v_i$ on processor $p_j$. When a task is affected to a processor $p_j$, it has to be the one with the highest priority in the $S$ matrix for $p_j$ over all the tasks that are ready to be executed. We assume that a single bucket is assigned to each type of task. As explained in "Heteroprio overview", this is not necessarily the case. The number of task types is expected to be significantly smaller than the total number of tasks. Thanks to this, our algorithms have complexity tied to $q$ or $b$ (rather than $v$) and run fast in practice. This can be illustrated by comparing the possible Heteroprio schedules against all the possible schedules. Consider a graph of 32 tasks with no edges (no dependencies), two different types (A and B), and one processor. As only the execution order of the tasks can change, there are $\binom{32}{1} = 32! \approx 2.63 \cdot 10^{35}$ possible schedules. The scheduling decisions that Heteroprio can take depend on the matrix $S$, which has only two possible configurations in this case: one where A has the highest priority and one where it is B. In every situation, Heteroprio has always exactly $(b!)^q$ possible schedules, where $b$ is the number of different task types, which is assumed to equal the number of buckets. As Heteroprio is designed to handle two processor types, we can simplify some notations. If $arch$ refers to the CPU, $\overline{arch}$ refers to the GPU and *vice versa*. It should be noted, however, that the heuristics have been generalized to more than two processing unit types. Additionally, $w_i^{arch}$ refers to the estimated cost of executing $v_i$ on processor $arch$. Finally, the presented model does not take into account memory transfers, as they are only to a small degree taken into account in Heteroprio.

## HEURISTICS FOR AUTOMATIC CONFIGURATION

In this section, we first detail the metrics that we use as a basis for our heuristics (Relevant metrics). The heuristics are described in a second step in "Heuristics for task prioritizing".

### Relevant metrics

We recall that we do not try to obtain priorities for each task but for each type of task. Consequently, when the number of predecessors, the number of successors, or the execution time are required, the average of all tasks of the same type is used. We also

**Table 1 Example of tasks costs and time differences for two types of tasks and two types of processors.**

| Task Worker | A | B |
|---|---|---|
| CPU | 100 s | 1 s |
| GPU | 130 s | 10 s |
| Relative difference ($w_i^{GPU}/w_i^{CPU}$) | ×1.3 | ×10 |
| Absolute difference ($w_i^{GPU} - w_i^{CPU}$) | 30 s | 9 s |

emphasize that these metrics are not the heuristics, but rather the values that are fed to the heuristics. These only aim at giving a quantitative input to the heuristics.

### CPU-GPU execution time difference

The CPU-GPU execution time difference can be expressed either as a relative or an absolute difference.

We use the following notations when referring to these metrics:

$$diff_{arch}(v_i) = w_i^{\overline{arch}} - w_i^{arch}, \tag{4}$$

$$rel\_diff_{arch}(v_i) = \frac{w_i^{\overline{arch}}}{w_i^{arch}}, \tag{5}$$

where $w_i^{arch}$ is the cost of $v_i$ on $arch$.

The idea of using these metrics is to be able to favor the most efficient architecture. Although the two metrics aim at measuring the same effect, they are not equivalent, as explained in the following example.

Let us consider the costs of two tasks on two architectures of Table 1. The question is which task type should a CPU worker favor. Here, we consider that both types of processors can execute tasks of types A and B. The relative difference would suggest executing B is a better choice, as its relative difference is higher (the CPU is 10 times faster). However, the absolute difference would suggest that A is a better choice, as it saves 30 s instead of 9 s.

The absolute and relative differences can, therefore, induce different scheduling choices.

### Normalized out-degree (NOD)

The normalized out-degree formula (*Lin et al., 2019*) is given by:

$$NOD(v_i) = \sum_{v_j \in succ(v_i)} \frac{1}{ID(v_j))}, \tag{6}$$

where $ID(v_j))$ is the inner degree of task $v_j$ (*i.e.*, its number of predecessors). This metric gives an indication about how many tasks can be expected to be released. In this view, it would mean that releasing $\frac{1}{ID(v_j))}$ of a task $v_j$ is as if it is partially released, at a proportion of $\frac{1}{ID(v_j))}$. For example, releasing two tasks at a "ratio" of $\frac{1}{2}$ can be viewed as being equivalent

to releasing 10 tasks at a ratio of $\frac{1}{10}$. This obscures the combinatorial nature of task-based execution but is a useful tool for guiding heuristics.

However, the NOD does not take into account the type of the tasks that will be released, which is critical in some cases. For example, in a case where we lack GPU jobs (starvation), the released GPU work is more beneficial than the released CPU work.

### Normalized released time (NRT)

We introduce the normalized released time (NRT). This metric is derived from the NOD and given by:

$$NRT_{arch}(v_i) = \sum_{v_j \in succ(v_i)} \frac{P_{exec}(v_j, arch) \cdot w_j^{arch}}{ID(v_j))}, \tag{7}$$

where $P_{exec}(v_j, arch)$ is the probability that $v_j$ is executed on architecture $arch$. This probability is not known during an execution. We instead measure the processor execution proportion of each task type during the execution and use this proportion as an approximation of the probability in our formula.

This formula is more refined than the first NOD formula for two reasons. Firstly, it takes into account the cost of the potential released successors. It is presumably better to release $N$ tasks with a cost of 10 s, than $N$ tasks of 1 s because it may release a higher workload. Secondly, CPU and GPU execution times are differentiated. This difference is crucial in a heterogeneous system. Having an NRT formula for both CPU and GPU gives information about where the released work is likely to be executed.

### Useful released time (URT)

We extend the normalized released time to define the useful released time given by:

$$URT(v_i) = NRT_{CPU}(v_i) \cdot IDLE(CPU) + NRT_{GPU}(v_i) \cdot IDLE(GPU), \tag{8}$$

where $IDLE(arch)$ is the idle proportion of $arch$ workers over all the execution. The URT represents how much useful time will be released after a task has finished its execution. The *useful time* is defined as the amount of released work that could help feeding the starving processors. This *useful released time* is estimated by scaling the released work (NRT) of each architecture to the idle proportion of the corresponding architecture. It is implied that the idle proportion is a relevant way of quantifying how much a processor is starving.

## Heuristics for task prioritizing

In this section, we present six heuristics: PRWS, PURWS, offset model, softplus model, interpolation model, and NOD-time combination[1].

### Parallel released work per second

In a typical scenario, tasks with high NOD scores should be encouraged to be executed as soon as possible, since they tend to release new tasks in the long run. In both theoretical and practical scenarios, however, using NOD alone as a score does not produce

[1] Other heuristics are presented in a research report (*Flint & Bramas, 2020*).

efficient priorities. Indeed, a task can have numerous successors (high NOD) but of low cost. If the costs of the successors are low, the newly released workload will also be low.

To take this effect into account, we introduce a new variable that is designed to give information about the quality of the released tasks. The idea is to keep a high degree of parallelism. This variable is the sum of the execution times of the successors of a task on their best architecture. With this variable we create the formula for the PRWS heuristic:

$$PRWS_{arch}(v_i) = \frac{NOD(v_i)}{w_i^{arch}} \cdot \left( \sum_{v_j \in succ(v_i)} \min_{arch \in Q} (w_j^{arch}) \right) + diff_{arch}(v_i) \tag{9}$$

Dividing by the cost of the task lets us measure the "releasing speed" (the released work comes at the cost of executing $v_i$). Adding $diff_{arch}(v_i)$ to the sum helps favoring the best architecture. To improve the work balance between the CPUs and the GPUs, the URT metric can be used instead of the NOD. The Eq. (9) becomes:

$$PURWS_{arch}(v_i) = \frac{URT(v_i)}{w_i^{arch}} \cdot \left( \sum_{v_j \in succ(v_i)} \min_{arch \in Q} (w_j^{arch}) \right) + diff_{arch}(v_i). \tag{10}$$

### Offset model
The offset model has a score that is defined by the following formula:

$$offset\_model_{arch}(v_i) = (URT(v_i) + \alpha) \cdot (diff_{arch}(v_i) + \beta). \tag{11}$$

In this model, the score is computed by multiplying $URT(v_i)$ and $diff_{arch}(v_i)$. $\alpha$ and $\beta$ are two hyperparameters that control the displacement for each of the two values. For example, if $\alpha = 0$ and $\beta = 0$, then tasks that have a $URT$ of 0 and those that have a $diff$ of 0 would have the same score (0), implying that they are equivalent in terms of criticality. The default values for $\alpha$ and $\beta$ are 1.3 and 1. This model has the downside of requiring two hyperparameters. Moreover, it is unable to distinguish between tasks when their $diff$ equals $-\beta$, even if their $URT$ are different.

### Softplus model
The softplus model is given by the formula:

$$softplus\_model_{arch}(v_i) = (1 + URT(v_i)) \cdot ln(1 + e^{diff_{arch}(v_i)}) \tag{12}$$

The idea of this model is comparable to that of the offset model but uses the *softplus* function ($softplus(x) = ln(1 + e^x)$). In contrary to the offset model, we multiply by *softplus* ($diff_{arch}(v_i)$) rather than by $diff_{arch}(v_i)$ directly. The *softplus* mostly changes the behavior of the heuristic when the *diff* is negative or around zero. This tends to negate the impact of *diff* when it tends towards zero.

### Interpolation model

The interpolation model combines the two previous models. When the *URT* approaches zero, it tends towards the offset model. It behaves more like the softplus model as the *URT* grows. It is given by:

$$interpolation\_model_{arch}(v_i) =$$
$$rpg(URT(v_i)) \cdot (1 + URT(v_i)) \cdot (1 + arch(v_i)) \qquad (13)$$
$$+(1 - rpg(URT(v_i))) \cdot ((-log(1 + exp(-archDiff)))),$$

where the interpolation is defined by the *rpg* function as follows:

$$rpg(x) = \begin{pmatrix} 1 & \text{if } x \geq 1 \\ \sqrt{x} \cdot \sqrt{2-x} & \text{otherwise} \end{pmatrix} \qquad (14)$$

This model aims at improving the two previous ones. We assume that the offset model gives particularly good priorities when *URT* is low and conversely for the softplus model. The idea is to perform an interpolation between the two models depending on the *URT* value and is controlled by the *rpg* function. $rpg(URT(v_i)) \in [0,1]$ because the *URT* is always positive. When $URT(v_i) = 0$, the interpolation model behaves like the offset model (with $\alpha = 1$ and $\beta = 1$). When $URT(v_i) \geq 1$, it behaves like the softplus model, but without the $(1 + URT(v_i))$ term.

### NOD-time combination

The NOD-time combination (NTC) heuristic is defined by the following formula:

$$NTC_{arch}(v_i) = diff_{arch} + \alpha \cdot NOD(v_i) \cdot e^{-\beta \cdot max\_rel\_diff^2} \qquad (15)$$

where

$$max\_rel\_diff = max(rel\_diff_{arch}(v_i), 1/rel\_diff_{arch}(v_i)) \qquad (16)$$

This equation needs two hyperparameters $\alpha$ and $\beta$. This heuristic aims at diminishing the importance of the NOD as the relative cost difference increases. $\alpha$ controls the importance of the *NOD*, compared to that of the *diff*, while $\beta$ controls the range in which the *NOD* is taken into account. If $rel\_diff_{arch}(v_i)$ is too high, the exponential is negated and the score equals $diff_{arch}$. The default value of $\alpha$ and $\beta$ are 0.3 and 0.5.

## Notes concerning the implementation in StarPU

### Cost normalization

If all the costs of the nodes of a DAG are scaled by a factor $\alpha$, the heuristics should give the same priorities. This would not be the case if we directly input the raw task costs. We, therefore, choose to normalize the costs of the task types.

Normalizing a set of heterogeneous costs is not straightforward. We propose the following normalization formula:

$$z_{i,j} = v \cdot \frac{w_{i,j}}{\sum_{0 \leq i < v} \min_{0 \leq j < q} (w_{i,j})}, \tag{17}$$

where $z_{i,j}$ is the normalized cost.

This formula normalizes the costs so that the average cost of a task on its best architecture equals 1. This method relies on the assumption that tasks are usually executed on their best architecture. This assumption, however, is disputable in some scenarios.

### Execution time prediction

The heuristics presented in this study rely on the execution times of the tasks. We consider that every task of a certain type has the same execution time. In practice, however, tasks of the same type can have radically different costs. Since the tasks have not been computed at the time they are pushed in the scheduler, we need to estimate their duration in real-time. We choose to approximate the cost of a task group by taking the average effective execution time of previous tasks of the corresponding type. If a task has never been executed on an architecture, we have no precise estimation of its execution time. We, therefore, implement two behaviors:

- the estimation is set to a default value of 100,000 s (default behavior);
- if an estimation exists on another processor, we take the fastest estimation, else we take 100,000 s.

This solution is imperfect, in particular when their execution times are dispersed. In this case, the scores given by the heuristics may translate into inefficient priorities. We assume that in most cases, taking the average execution time is sufficient for generating reliable priorities.

### Task-graph

In this model, we consider that the applications are converted into a task graph which is a DAG. Most memory access types (READ, WRITE, READ-WRITE) can be translated in a dependency in a DAG. Some accesses, however, cannot be transcribed in terms of direct static dependencies. For example, StarPU has a memory access type known as STARPU_COMMUTE which is used when several contiguous (READ-)WRITE accesses can be performed in any order but not at the same time. A simplistic use case of this would be when the tasks increment a shared counter. This access mode has been used in mathematical applications, *e.g.*, for an optimized discontinuous Galerkin solver or the fast multipole method (*Agullo et al., 2017*; *Bramas et al., 2020*). For this type of access, we can reason in terms of availability rather than dependency: (1) if no task is commuting on the data, any task can take the memory node, and (2) if one task is commuting, the memory node is blocked. Thus, the heuristics cannot use all the information they have on applications that use these relatively uncommon memory access types. In practice, in the presented heuristics, these accesses are treated as write accesses.

*Heteroprio automatic configuration*

In our implementation, we update the priority lists in the scheduler only when a task is pushed in the scheduler. More precisely, the priorities are updated the first time a task is pushed (the first time the scheduler discovers a new type of task), and then every $n^{th}$ pushed task. This choice avoids updating the priorities too often and should, therefore, help reduce the scheduling overhead.

# PERFORMANCE STUDY

## Evaluation based on emulated executions

We create a simple simulator for running a fake StarPU execution. As input, it takes the fake DAG of an application, the costs of the tasks, and the priority lists. It then simulates an execution with the Heteroprio scheduler based on our model (see "Heteroprio automatic configuration problem"). As output, it gives the theoretical execution time of the whole fake application. This theoretical execution time does not include data transfers.

It can be viewed as a black box where we input priorities and obtain an execution time as output. We, therefore, choose this tool as a base for elaborating our experimental protocol. This protocol aims at generating a score for a heuristic based on how well it performs in multiple scenarios. It has two purposes. Firstly, it provides a fast way to check how successful a heuristic is. Secondly, it provides an additional argument for our work if the heuristics perform as well in the protocol as in real applications.

*Graph generation*

To be able to evaluate our heuristics, we generate a dataset of 32 graphs with diversity in the number of task types, the costs of the tasks, and the graph shape. To generate a graph, we generate tasks while filling a pipeline of workers[2]. We affect each task to its best worker. Consequently, at the end of the generation process, we know the scheduling that minimizes the makespan and have a lower bound for a hardware configuration that corresponds to the pipeline. We also generate a predecessor matrix $P$ randomly. This predecessor matrix is of size $v \times v$ and $P_{i,j}$. It represents the average number of predecessors of tasks of type $i$ that are of type $j$. Our graph generation method uses this predecessor matrix as input and adjusts the predecessors of the newly created tasks so that they match the values of the matrix.

The generator needs the following parameters:

- a seed for the generation of random numbers
- the final amount of tasks
- a list of task types, with their associated CPU and GPU costs and their expected proportion in the pool of tasks
- a number of CPU and GPU workers
- a predecessor matrix

Table 2 gives details about the generated datasets.

[2] The DAG generating code is publicly available (*Bramas, Flint & Paillat, 2021*).

**Table 2 Details of the graph dataset generated randomly.** CPU-GPU close tasks are the tasks that have less than +20% between the two processor costs and conversely for far CPU-GPU costs. Here, "numerous" means five or more.

| Data index | CPU number | GPU number | CPU/ GPU | Close CPU-GPU task proportion | Far CPU-GPU task proportion | Task with numerous predecessors proportion | Average predecessor number | Max predecessor number | Task without successor proportion | Task with numerous successors proportion | Max successor number | Average CPU-GPU diff (relative) |
|---|---|---|---|---|---|---|---|---|---|---|---|---|
| 0 | 4 | 14 | 0.286 | 0.549 | 0.336 | 0.502 | 4.038 | 7 | 0.423 | 0.243 | 41 | 0.580 |
| 1 | 13 | 8 | 1.625 | 0.377 | 0.623 | 0.000 | 2.101 | 3 | 0.467 | 0.084 | 105 | 0.905 |
| 2 | 11 | 12 | 0.917 | 0.687 | 0.135 | 0.000 | 2.526 | 3 | 0.781 | 0.033 | 540 | 0.855 |
| 3 | 2 | 7 | 0.286 | 0.191 | 0.302 | 0.211 | 2.529 | 4 | 0.388 | 0.104 | 142 | 1.089 |
| 4 | 13 | 15 | 0.867 | 0.508 | 0.233 | 0.007 | 1.289 | 5 | 0.575 | 0.057 | 102 | 1.605 |
| 5 | 9 | 7 | 1.286 | 0.276 | 0.573 | 0.141 | 2.301 | 4 | 0.360 | 0.127 | 546 | 1.154 |
| 6 | 13 | 3 | 4.333 | 0.314 | 0.026 | 0.000 | 2.396 | 3 | 0.339 | 0.118 | 62 | 0.715 |
| 7 | 11 | 1 | 11.000 | 0.226 | 0.531 | 0.000 | 1.491 | 3 | 0.496 | 0.062 | 307 | 1.036 |
| 8 | 9 | 12 | 0.750 | 0.418 | 0.582 | 0.000 | 1.675 | 3 | 0.255 | 0.070 | 22 | 0.187 |
| 9 | 12 | 1 | 12.000 | 0.405 | 0.043 | 0.367 | 2.927 | 4 | 0.176 | 0.180 | 57 | 0.388 |
| 10 | 2 | 9 | 0.222 | 0.167 | 0.777 | 0.000 | 0.995 | 1 | 0.301 | 0.005 | 7 | 3.791 |
| 11 | 13 | 10 | 1.300 | 0.232 | 0.529 | 0.000 | 1.384 | 3 | 0.507 | 0.083 | 24 | 1.720 |
| 12 | 4 | 6 | 0.667 | 0.286 | 0.530 | 0.000 | 1.325 | 3 | 0.658 | 0.060 | 103 | 1.179 |
| 13 | 4 | 11 | 0.364 | 0.018 | 0.497 | 0.000 | 1.462 | 3 | 0.563 | 0.031 | 72 | 1.484 |
| 14 | 8 | 1 | 8.000 | 0.498 | 0.468 | 0.000 | 1.850 | 2 | 0.459 | 0.088 | 52 | 1.756 |
| 15 | 3 | 8 | 0.375 | 0.112 | 0.888 | 0.000 | 2.686 | 3 | 0.570 | 0.072 | 111 | 4.153 |
| 16 | 15 | 3 | 5.000 | 0.294 | 0.126 | 0.000 | 1.347 | 2 | 0.466 | 0.094 | 20 | 1.243 |
| 17 | 10 | 1 | 10.000 | 0.452 | 0.514 | 0.000 | 2.258 | 3 | 0.766 | 0.064 | 548 | 0.228 |
| 18 | 7 | 3 | 2.333 | 0.160 | 0.565 | 0.139 | 1.679 | 4 | 0.432 | 0.094 | 54 | 1.793 |
| 19 | 9 | 14 | 0.643 | 0.269 | 0.725 | 0.000 | 1.817 | 3 | 0.334 | 0.084 | 108 | 2.238 |
| 20 | 8 | 11 | 0.727 | 0.386 | 0.392 | 0.000 | 1.859 | 3 | 0.294 | 0.093 | 25 | 0.850 |
| 21 | 8 | 8 | 1.000 | 0.527 | 0.324 | 0.323 | 2.655 | 5 | 0.386 | 0.083 | 439 | 0.917 |
| 22 | 15 | 9 | 1.667 | 0.350 | 0.650 | 0.126 | 2.268 | 4 | 0.281 | 0.107 | 147 | 2.050 |
| 23 | 14 | 4 | 3.500 | 0.008 | 0.973 | 0.000 | 1.288 | 3 | 0.228 | 0.022 | 116 | 12.786 |
| 24 | 1 | 2 | 0.500 | 0.115 | 0.175 | 0.133 | 1.934 | 5 | 0.327 | 0.115 | 18 | 0.881 |
| 25 | 9 | 11 | 0.818 | 0.278 | 0.278 | 0.000 | 2.030 | 3 | 0.275 | 0.119 | 13 | 0.626 |
| 26 | 4 | 14 | 0.286 | 0.299 | 0.512 | 0.166 | 1.884 | 4 | 0.372 | 0.111 | 34 | 0.771 |
| 27 | 15 | 1 | 15.000 | 0.453 | 0.417 | 0.000 | 1.551 | 2 | 0.685 | 0.090 | 55 | 0.253 |
| 28 | 9 | 3 | 3.000 | 0.635 | 0.266 | 0.099 | 1.474 | 5 | 0.477 | 0.066 | 131 | 0.187 |
| 29 | 15 | 8 | 1.875 | 0.288 | 0.539 | 0.396 | 3.558 | 6 | 0.186 | 0.264 | 50 | 1.534 |
| 30 | 10 | 10 | 1.000 | 0.612 | 0.000 | 0.169 | 2.552 | 7 | 0.368 | 0.197 | 28 | 0.434 |
| 31 | 12 | 13 | 0.923 | 0.395 | 0.605 | 0.000 | 1.516 | 3 | 0.482 | 0.094 | 17 | 2.245 |

*Protocol*

We run fake executions on the 32 generated graphs for each heuristic. We compare the obtained makespans to the makespans obtained with control priorities and provide a slowdown for each heuristic. The control priorities are obtained with an iterative optimization algorithm. The algorithm begins with random CPU and GPU priorities. It then performs multiple iterations, alternating between CPU and GPU. At every iteration, all the possible priority permutations for the current architecture (CPU/GPU) are tested

Table 3 **Slowdown obtained on emulated executions by comparing the estimated lower-bound against Heteroprio-based executions using the different heuristics.** The lower bound is estimated with an iterative optimization algorithm.

| Heuristic Test case | Offset | Softplus | Interpolation | PURWS | PRWS | NTC |
|---|---|---|---|---|---|---|
| 0 | 1.119 | 1.120 | 1.119 | 1.285 | 1.285 | 1.135 |
| 1 | 1.001 | 1.049 | 1.001 | 1.062 | 1.062 | 1.001 |
| 2 | 1.045 | 1.031 | 1.063 | 1.170 | 1.264 | 1.209 |
| 3 | 1.096 | 1.154 | 1.032 | 1.178 | 1.208 | 1.241 |
| 4 | 1.117 | 1.104 | 1.138 | 1.159 | 1.119 | 1.110 |
| 5 | 1.052 | 1.048 | 1.007 | 1.194 | 1.165 | 1.062 |
| 6 | 1.318 | 1.280 | 1.361 | 1.182 | 1.061 | 1.452 |
| 7 | 1.436 | 1.536 | 1.530 | 1.752 | 1.056 | 1.019 |
| 8 | 1.144 | 1.078 | 1.076 | 1.046 | 1.017 | 1.025 |
| 9 | 1.029 | 1.042 | 1.029 | 1.017 | 1.017 | 1.023 |
| 10 | 1.329 | 1.329 | 1.329 | 1.000 | 1.000 | 1.048 |
| 11 | 1.010 | 1.010 | 1.010 | 1.128 | 1.160 | 1.010 |
| 12 | 0.992 | 1.009 | 1.035 | 1.038 | 1.047 | 0.990 |
| 13 | 1.026 | 1.126 | 1.069 | 1.183 | 1.183 | 1.126 |
| 14 | 1.014 | 1.003 | 1.014 | 1.034 | 1.034 | 1.014 |
| 15 | 1.010 | 1.010 | 1.010 | 1.321 | 1.281 | 1.283 |
| 16 | 1.020 | 1.297 | 1.297 | 1.020 | 1.020 | 1.020 |
| 17 | 1.052 | 1.019 | 1.058 | 1.050 | 1.050 | 1.026 |
| 18 | 1.193 | 1.054 | 1.040 | 1.366 | 1.304 | 1.193 |
| 19 | 1.163 | 1.487 | 1.163 | 1.224 | 1.279 | 1.354 |
| 20 | 1.000 | 1.000 | 1.268 | 1.254 | 1.254 | 1.163 |
| 21 | 1.156 | 1.251 | 1.156 | 1.474 | 1.351 | 1.158 |
| 22 | 1.134 | 1.118 | 1.134 | 1.143 | 1.197 | 1.065 |
| 23 | 0.999 | 1.126 | 0.999 | 1.002 | 1.126 | 1.154 |
| 24 | 1.007 | 1.055 | 1.020 | 1.191 | 1.154 | 1.043 |
| 25 | 1.042 | 1.068 | 1.042 | 1.037 | 1.037 | 1.055 |
| 26 | 1.124 | 1.063 | 1.076 | 1.075 | 1.084 | 1.070 |
| 27 | 1.028 | 1.028 | 1.013 | 1.002 | 1.002 | 1.019 |
| 28 | 1.034 | 1.018 | 1.114 | 1.065 | 1.077 | 1.034 |
| 29 | 1.092 | 1.082 | 1.083 | 1.159 | 1.159 | 1.009 |
| 30 | 1.014 | 1.014 | 1.014 | 1.075 | 1.051 | 0.992 |
| 31 | 1.106 | 1.106 | 1.106 | 1.118 | 1.118 | 1.118 |

and the fastest permutation is kept. In the case of a tie, the fastest priorities are chosen randomly among the equally-ranked bests. These control priorities aim at giving anchor points for computing the slowdowns of the heuristics.

### Results

The results of our emulated simulations are available in Table 3. We show the slowdown of the six heuristics we present in this article (compared to the control priorities): PRWS,

PURWS, offset model, softplus model, and interpolation model. We see that some slowdowns are lower than 1. This means that some heuristics find better priorities than the control priorities, which have been found with an iterative optimization algorithm. In general, we observe that the slowdown ranges between +0% and +20%. In most test cases, the best of the six heuristics usually has a slowdown of less than +10%. There are some exceptions such as cases number 0, 4, 19, 21, 22, or 31. From these simulated executions, we expect the choice of heuristic to have a significant impact.

## Evaluation on real applications

### Configuration

#### Hardware

We carry out our experiments on three configurations. Each one has a different GPU model. In this article, we use the model name of the GPUs for referring to the associated configuration:

- **K40M** is composed of 2 Dodeca-cores Haswell Intel Xeon E5-2680 v3 2.5 GHz, and 4 K40m GPUs (4.29 TeraFLOPS per GPU). We use 7 CUDA streams per GPU;
- **P100** is composed of 2 Hexadeca-core Broadwell Intel Xeon E5-2683 v4 2.1 GHz, and 2 P100 GPUs (8.07 TeraFLOPS per GPU). We use 16 CUDA streams per GPU;
- **V100** is composed of 2 Hexadeca-core Skylake Intel Xeon Gold 6142 2.6 GHz, and 2 V100 GPUs (14.0 TeraFLOPS per GPU). We use 16 CUDA streams per GPU.

#### Software

We select four applications that are already parallelized with StarPU to evaluate our scheduler:

- **ScalFMM** (*Agullo et al., 2014*) is an application that implements the fast multipole method (FMM). The FMM algorithm computes the n-body interactions between the particles directly and across a tree mapped over the simulation box. We use it with two test-cases based on the *testBlockedRotationCuda* program. The first one runs with the default parameters and 10 million particles. The other one runs with a block size of 2,000, a tree height of 7, and 60 million particles;
- **QrMUMPS** computes the QR factorization of sparse matrices (*Agullo et al., 2013*) using the multifrontal method (*Duff & Reid, 1983*). When it was extended to heterogeneous architectures in 2016 by Florent LOPEZ (*Lopez, 2015*), Heteroprio was the fastest scheduler of StarPU for this application. In our experiment, we choose to measure the factorization time of the TF16 matrix (*Thiery, 2008*), from the JGD_Forest dataset;
- **Chameleon** is a library for dense linear algebra operations that supports heterogeneous architectures (*Agullo et al., 2010*). We select the same operations as the ones considered by the authors for the benchmarks presented in their user guide: a Matrix Multiplication (GEMM), a QR factorization (QRM), and a Cholesky factorization (POTRF). We use a block size of 1,600 and a matrix size of 40,000 for the Matrix
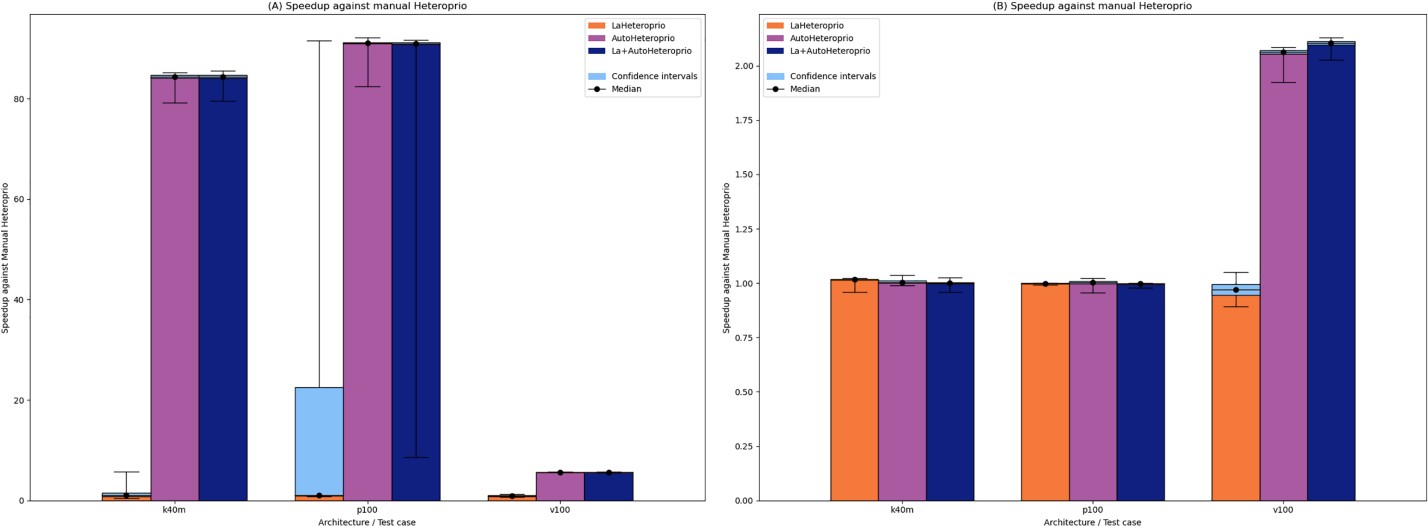

**Figure 2** **Speedups of LaHeteroprio, AutoHeteroprio, and LaAutoHeteroprio against Heteroprio in the two ScalFMM test cases.** We study two test cases: (A) Ten million particles and (B) 60 million particles. The hatched area represents the interval of confidence of the 32 corresponding runs.

Multiplication and the QR factorization. For the Cholesky factorization, we use a matrix size of 50,000;

- **PaStiX** is a library which provides a high performance solver for sparse linear systems (*Hénon, Ramet & Roman, 2002*; *Lacoste, 2015*). We consider two stages of the example program named 'simple': the LU factorization and the solve step. The program generates a Laplacian matrix. We choose a matrix size of $100^3$.

For a given set of parameters (scheduler, hardware configuration, etc.), each application is run 32 times. All these applications can be configured to use StarPU and, therefore, the task-based model. The codelets (low-level kernels) are encapsulated into tasks that are submitted to StarPU. The four applications have CPU and CUDA kernels and at least one task that has both a CPU and a CUDA implementation. For the latter hybrid tasks, the scheduler is responsible for making the proper processor type choice. Finally, the tested applications are all written in C, except for QrMUMPS which is written in Fortran. To make the applications usable for our tests, we change parts of them. We update QrMUMPS and ScalFMM so that they use performance models, which are needed by our automatic strategy but also by most schedulers. Additionally, we create new static priorities for the Heteroprio scheduler in Chameleon and PaStiX. The methodology for setting these priorities is detailed in the "Manual priority settings". Unless otherwise indicated, the execution times are the median value of the 32 corresponding runs. All schedulers that need a calibration run (which sets up the performance models) use an extra run that is not included in the final results.

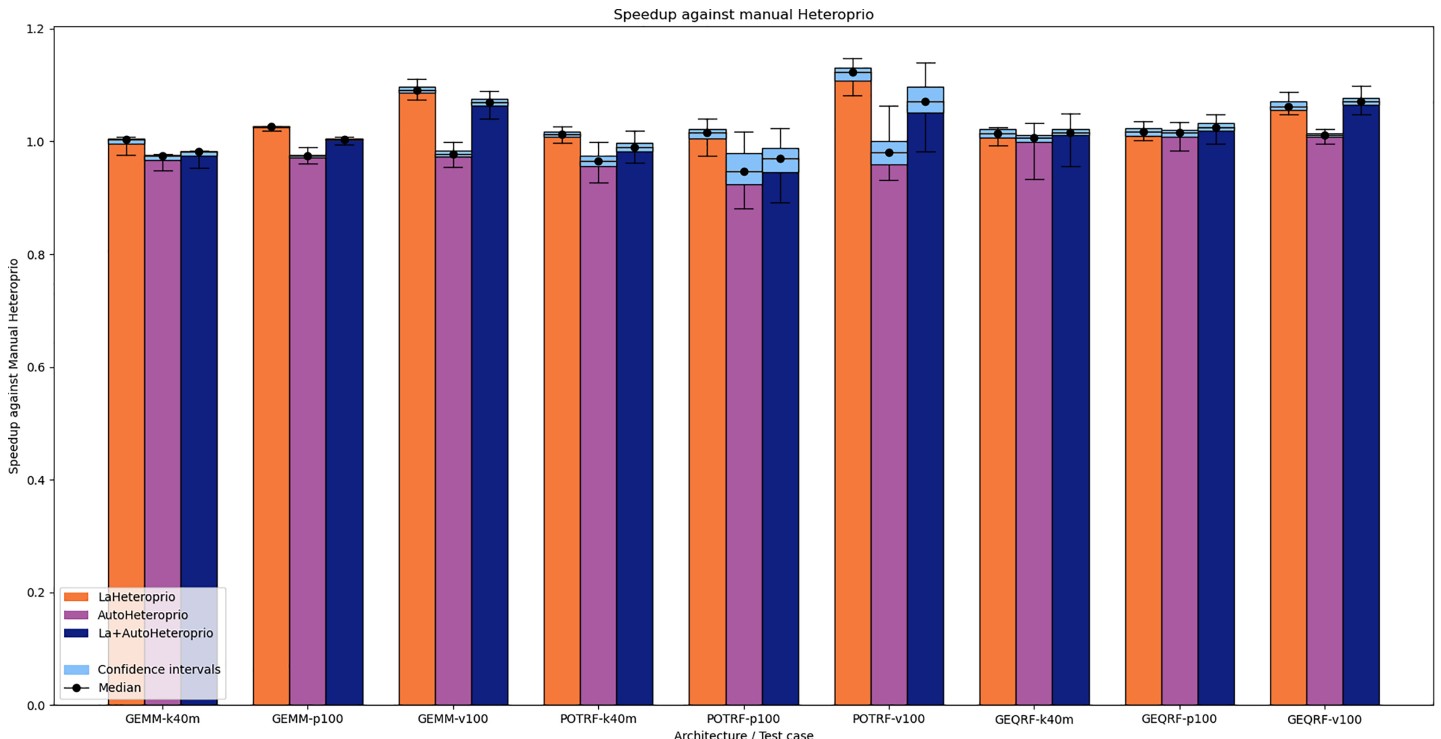

**Figure 3** **Speedups of LaHeteroprio, AutoHeteroprio, and LaAutoHeteroprio against Heteroprio on Chameleon test cases (GEMM, POTRF, and GEQRF).** P100, V100 and K40M relate to the hardware configuration. The hatched area represents the interval of confidence of the 32 corresponding runs.

### Comparison between manual and automatic priorities

In this study, we compare the performance of four versions of the scheduler: Heteroprio, LaHeteroprio, AutoHeteroprio, and LaAutoHeteroprio (AutoHeteroprio with LA enabled). We use Heteroprio as the reference value and provide the speedups of the three other versions. For AutoHeteroprio and LaAutoHeteroprio, we provide the data of the best heuristic, *i.e.*, the heuristic whose average execution time is the lowest. The median execution time of Heteroprio (the reference) is divided by each individual execution time for obtaining speedups. By doing so, we obtain a set of speedups for each case, rather than a single value. This lets us display a median and two limits of a confidence interval. For this confidence interval, we exclude the 5% highest and 5% lowest values.

Figure 2 shows the results for ScalFMM. In the first test case (10 million particles), all the versions are comparable on the p100 and k40m architecture. In the v100 case, AutoHeteroprio and LaAutoHeteroprio are about 2 times faster than normal Heteroprio. In the second test case (60 million particles), AutoHeteroprio and LaAutoHeteroprio are more than 80 times faster on the p100 and k40m architectures and about 5 times faster on the v100 architecture. LaHeteroprio (respectively, LaAutoHeteroprio) does not show such a high difference to Heteroprio (respectively, AutoHeteroprio) in this scenario. The reason for this is that data transfers are hard to avoid in this application because only

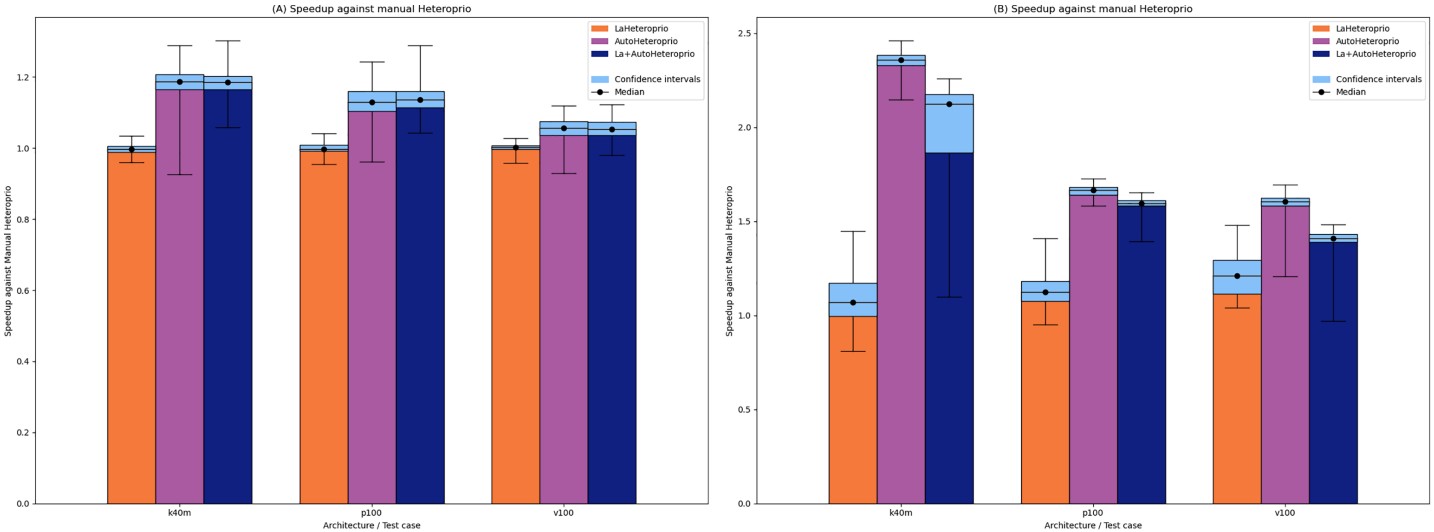

**Figure 4** **Speedups of LaHeteroprio, AutoHeteroprio, and LaAutoHeteroprio against Heteroprio in theQr factorization (QrMumps) and the LU factorization (PaStiX).** (A) Qr factorization (QrMUMPS). (B) LU factorization (PaStiX). The hatched area represents the interval of confidence of the 32 corresponding runs.

two task types have a GPU implementation. Their data must be transferred back to the main memory to be used by tasks on the CPU.

Figure 3 shows the results for Chameleon. In this application, automatic priorities are systematically slower than their manual counterparts. Indeed, AutoHeteroprio generally has a speedup of less than 1 and LaAutoHeteroprio is usually worse than LaHeteroprio. Furthermore, LaHeteroprio and LaAutoHeteroprio tend to be faster, which suggests that locality has greater importance in Chameleon than in ScalFMM.

We explain the lack of performance of automatic versions by a lack of precision in the execution time estimations of the tasks. This leads to an inefficient choice of priorities. The execution time estimations of the tasks are biased because AutoHeteroprio averages the execution time of a task type. Yet, in Chameleon, the data size has an important impact on the execution times of the tasks. This breaks our initial premise which is that each task within a bucket has the same execution time.

We provide the results for the QR Factorization from QrMUMPS and on the LU Factorization from PaStiX in Figs. 4A and 4B, respectively. In both cases, AutoHeteroprio shows a significant increase in performance on all configurations. In the QR-MUMPS test, AutoHeteroprio reaches more than +18% speedup. In the LU factorization, it goes past x2.3 speedup on the k40m architecture. It appears that the dynamic change of the priorities at runtime of the automatic Heteroprio is an advantage in both applications (to evaluate these changes, we manually export the priorities during the executions).

Overall, we have multiple observations. It appears that using automatic priorities does not always harm performance. In some cases, it can even increase them. Automatic priorities are only slower in the case of the GEMM and POTRF test cases in Chameleon. In some cases, the speedups of the automatic priorities become particularly high when run on

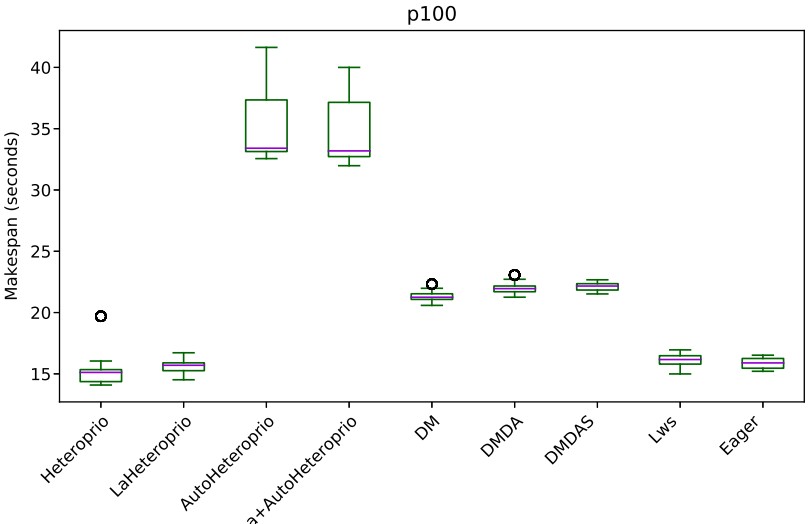

**Figure 5 Execution times of the PaStiX solve step for different schedulers on the P100 configuration.** The boxes show the distribution of the 32 makespans (896 for AutoHeteroprio and LaHeteroprio) for each scheduler.          

a new architecture (*e.g.*, Fig. 2). This demonstrates the ability of automatic priories to adapt to the current architecture. Manual priorities, on the other hand, can hardly be efficient on multiple different architectures.

### Comparison with other schedulers

In this section, we compare Heteroprio with other schedulers available in StarPU:

- the *Eager* scheduler uses a central task queue from which all workers retrieve tasks concurrently. There is no decision on the task distribution. The worker picks the first task that is compatible with their PU;
- the *LWS* (Locality Work Stealing) scheduler uses one queue per worker. When a task becomes ready, it is stored in the queue of the worker that released it. When the queue of a worker is empty, the worker tries to steal tasks from the queues of other workers;
- the *Random* scheduler randomly assigns the tasks to compatible workers;
- the *DM* (deque model) scheduler uses a HEFT-like strategy. It tries to minimize the makespan by using a look-ahead strategy;
- the *DMDA* (deque model data aware) follows the principle of DM but adds the data transfer costs;
- the *DMDAS* (deque model data aware) acts as the DMDA scheduler but lets the user affect priorities to the tasks. Since this scheduler needs user-defined priorities, we discard DMDAS from the results when the application does not define custom priorities.

For the sake of conciseness, by default we only display the results for the best between Heteroprio (respectively AutoHeteroprio) and LaHeteroprio (respectively LaAutoHeteroprio). When the difference between the LA and the non-LA version is noticeable, we display the four versions. For the automatic Heteroprio versions

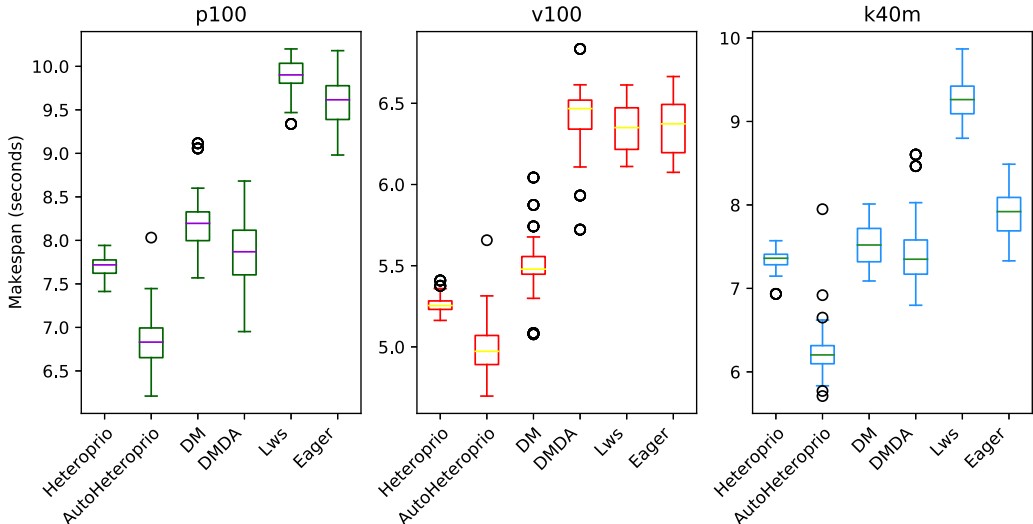

**Figure 6 Execution times for QrMumps for different schedulers on the three configurations.** The boxes show the distribution of the 32 makespans (896 for AutoHeteroprio) for each case.

(AutoHeteroprio and LaHeteroprio), we aggregate all the data of every heuristic designed in AutoHeteroprio. Since there are 28 different heuristics in AutoHeteroprio and 32 runs for each one, the data for the automatic configuration consists of $28 \times 32 = 896$ runs, while other the other shown data consist of 32 runs.

Figure 5 shows the execution times of the solve step in PaStiX with different schedulers on the p100 configuration (the results for the v100 and k40m configurations are comparable).

We can group the schedulers into three performance categories (sorted from slowest to fastest):

- AutoHeteroprio and LaAutoHeteroprio
- DM, DMDA, and DMDAS
- basic Heteroprio, LaHeteroprio, LWS, and Eager

To explain this result, let us explain the task structure of this application. There are only two types of tasks with average execution times of 95 and 120 microseconds. These execution times are relatively short for a runtime system like StarPU. Indeed, the overhead of StarPU is relatively high, as it has been designed to handle large amounts of data. In particular, the use of a scheduler is only relevant when the expected gained time is greater than the overhead of the scheduler. In this test case, it appears that the scheduling decision has less importance than in other applications, as lightweight schedulers tend to perform better. It confirms that Heteroprio and LaHeteroprio have a low overhead. Their overhead is comparable to those of LWS and Eager. This test also points out that AutoHeteroprio and AutoLaHeteroprio have a significant overhead. For these, the overhead is higher than that of DM, DMDA, and DMDAS.

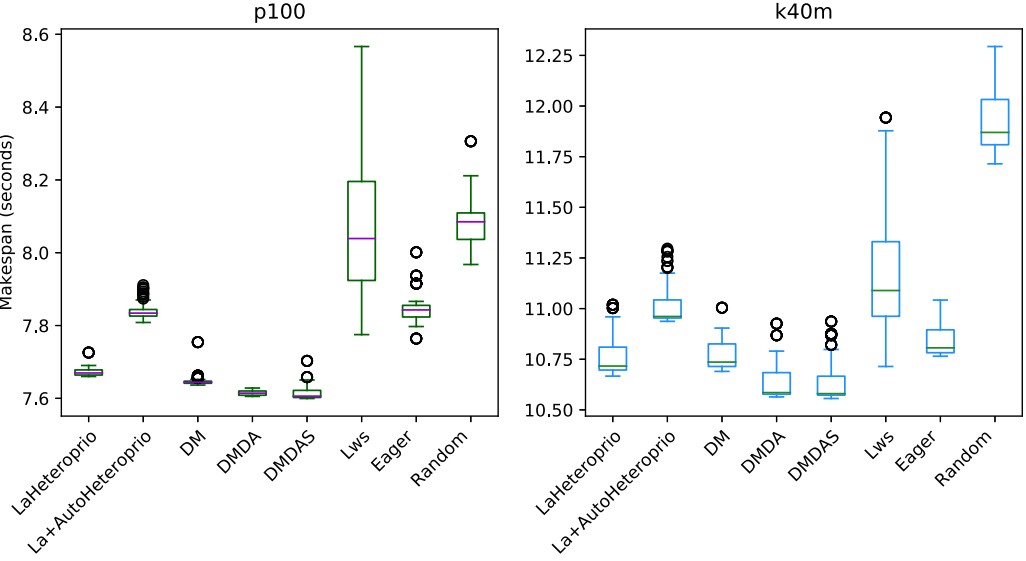

**Figure 7 Execution times for Chameleon GEMM for different schedulers on two configurations.** The boxes show the distribution of the 32 makespans (896 for LaAutoHeteroprio) for each case.

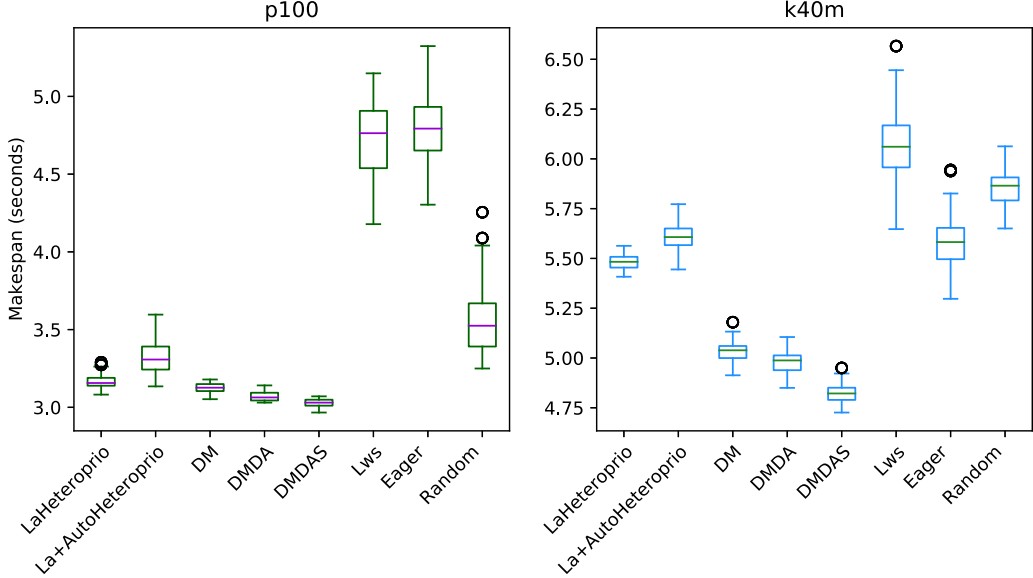

**Figure 8 Execution times for Chameleon Cholesky factorization for different schedulers on the p100 and the k40m configuration.** The boxes show the distribution of the 32 makespans (896 for LaAutoHeteroprio) for each scheduler.

We compare the schedulers for the QrMUMPS test case in Fig. 6. AutoHeteroprio performs better than manual Heteroprio, which is already better or as good as other schedulers, depending on the configuration.

Figure 7 presents the results for the Matrix multiplication in Chameleon, on the k40m and the p100 configurations. The V100 has been left out as the results are similar to the P100 configuration. We observe that AutoHeteroprio is faster and more reliable than

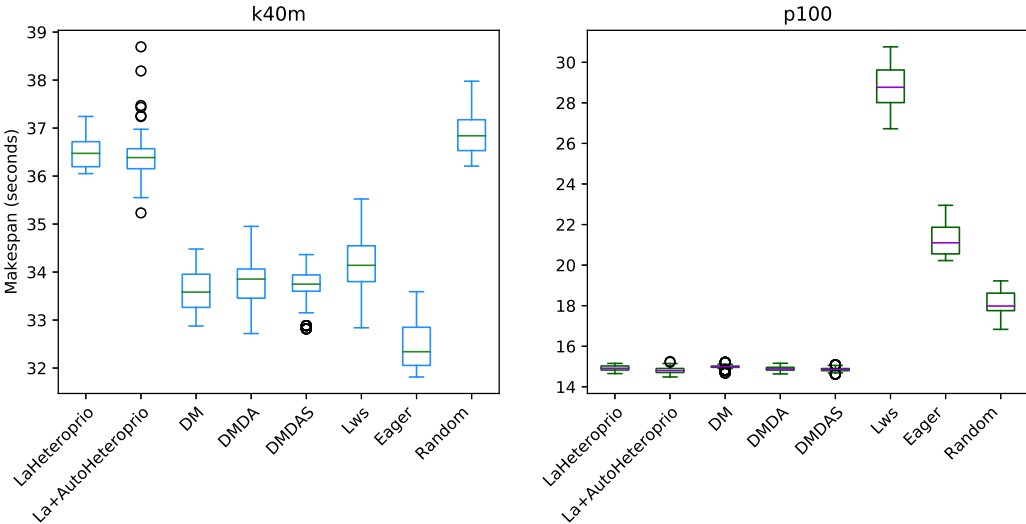

**Figure 9 Execution times for Chameleon QR factorization for different schedulers on the p100 and the k40m configuration.** The boxes show the distribution of the 32 makespans (896 for LaAutoHeteroprio) for each scheduler.

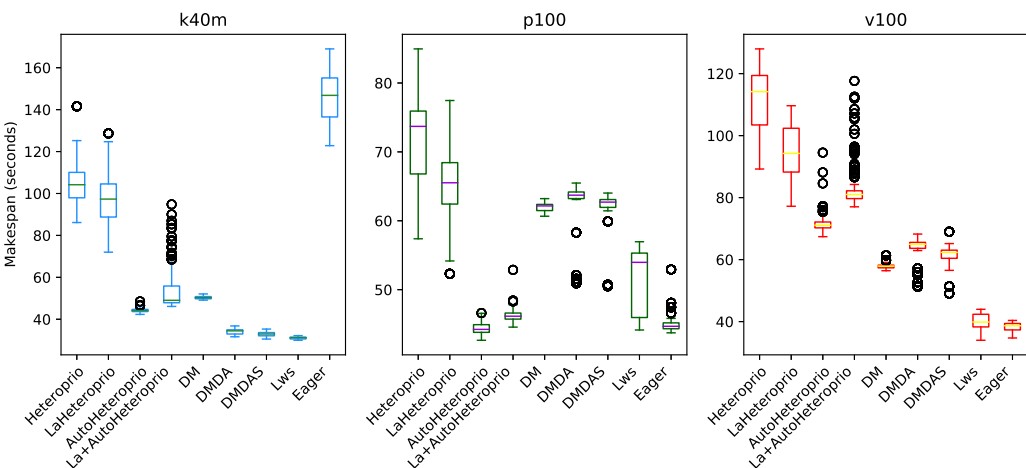

**Figure 10 Execution times for PaStiX factorization for different schedulers on the three hardware configurations (k40m, p100 and v100).** The boxes show the distribution of the 32 makespans (896 for AutoHeteroprio and LaAutoHeteroprio) for each scheduler.

schedulers like LWS or random but less efficient than the DM schedulers. The results for the Cholesky factorization that we present in Fig. 8, are similar. In this configuration, AutoHeteroprio is closer to the performances of DM. Manual Heteroprio performs almost as well as DM.

We present the results for the Chameleon QR Factorization in Fig. 9. In the p100 configuration (and the v100 configuration which is comparable), both Heteroprio versions perform comparably to the DM scheduler. In the k40m configuration, the performance of both versions is low. Heteroprio only seems to do better than the random scheduler. The Eager scheduler outmatches DM schedulers.

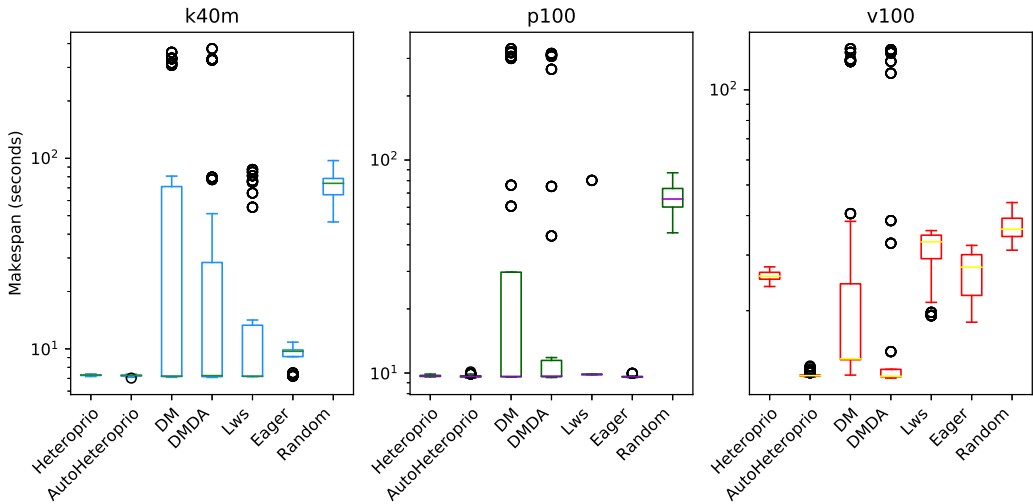

**Figure 11 Execution times for the first ScalFMM test case on the three hardware configurations (k40m, p100 and v100).** The scale of the Y-axis is logarithmic. The boxes show the distribution of the 32 makespans (896 for AutoHeteroprio) for each scheduler.

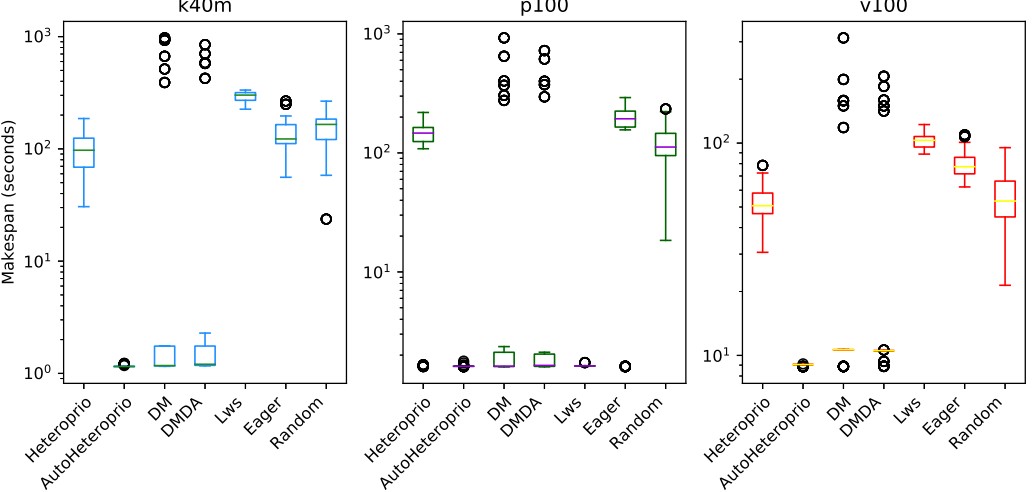

**Figure 12 Execution times for the second ScalFMM test case on the three hardware configurations (k40m, p100 and v100).** The scale of the Y-axis is logarithmic. The boxes show the distribution of the 32 makespans (896 for AutoHeteroprio) for each scheduler.

In the case of factorization with PaStiX (Fig. 10), AutoHeteroprio performs well on the p100 configuration. In contrast, on the k40m configuration, DMDAS, DMDA, and LWS schedulers perform better. With the v100 configuration, the results of AutoHeteroprio are only better than the ones of Heteroprio.

The results of the ScalFMM tests cases are shown in Figs. 11 and 12. These are represented using a logarithmic scale because of the high differences between the execution times of the schedulers. We can see that AutoHeteroprio performs well on this application. It is comparable and sometimes better than schedulers of the DM family. Note that the DM and DMDA schedulers can use more than one calibration run. This presumably

**Table 4 Longest and shortest relative time observed between heuristics across all test-cases.**

| Application | FMM (%) | Chameleon POTRF (%) | Chameleon GEMM (%) | Chameleon GEQRF (%) | PaStiX (%) | QrMUMPS (%) |
|---|---|---|---|---|---|---|
| Longest time | +3 | +5 | <+1 | <+1 | +1.5 | +1 |
| Shortest time | >−1 | −4 | >−1 | >−1 | −1 | −1 |

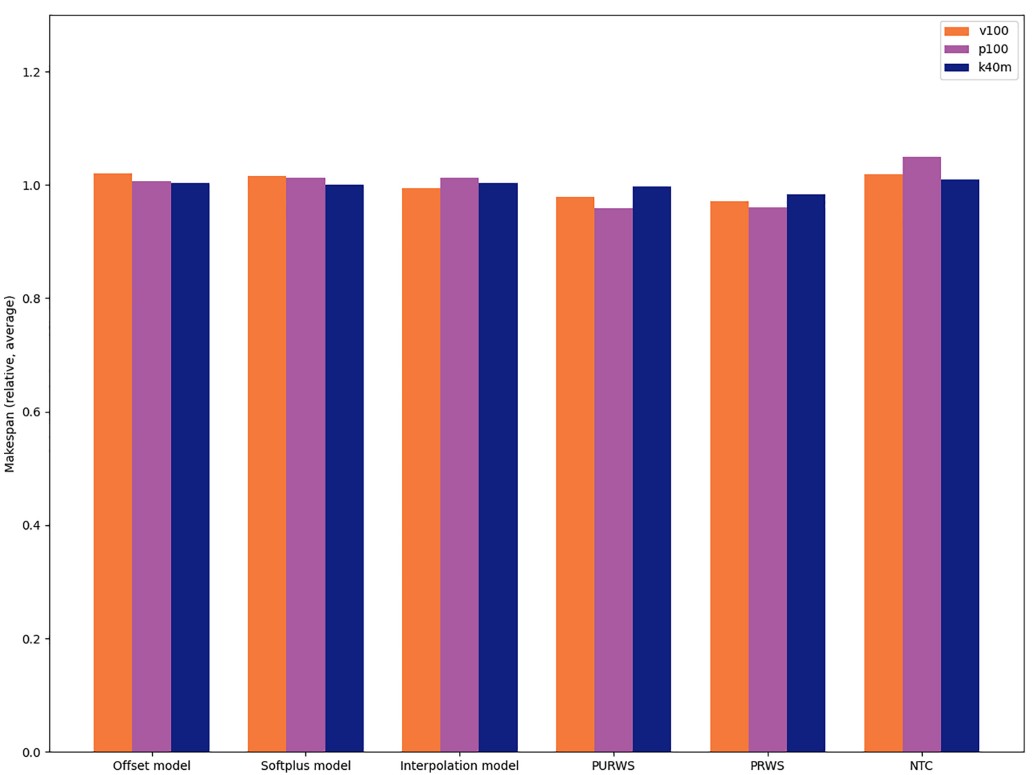

**Figure 13 Relative difference between six heuristics in the case of the Cholesky factorization (Chameleon POTRF).**

explains their uppermost bullets in the figures. AutoHeteroprio only needs one calibration run before achieving its best performance.

This second study gives an overview of the performance of different applications with various schedulers in StarPU. With these results we can estimate the impact of the choice of scheduler on the overall execution time and evaluate the competitiveness of Heteroprio with manual or automatic priorities. In general, AutoHeteroprio offers satisfying results compared to its competitors. When it does not, it is usually in cases where the Heteroprio (manual) version is already slow. The only cases where AutoHeteroprio does not achieve acceptable performance when compared to Heteroprio are the Chameleon GEMM and the PaStiX solve step. Moreover, AutoHeteroprio does improve the performance of Heteroprio significantly in other cases such as in QrMumps, PaStiX factorization, and some ScalFMM configurations. Therefore, this study suggests that AutoHeteroprio is a competitive scheduler for a runtime system like StarPU. In addition to

this, it is fully automatic, contrary to some of its competitors (Heteroprio, LaHeteroprio, and DMDAS).

### Comparison of different heuristics in AutoHeteroprio

In AutoHeteroprio, the priority lists are computed thanks to heuristics. In "Comparison between manual and automatic priorities", we show the performance of the best heuristic over all the 28 measured executions, while in "Comparison with other schedulers" we show the aggregated performance of the 28 heuristics. In this section we seek to measure the impact of the choice of heuristic. We compute the average execution time of each heuristic and compare it against the average execution time of all heuristics. We establish the results shown in Table 4, which are the maximum and minimum differences observed across all the 28 heuristics on each application. While it appears that the relative difference is relatively low, typically around 1%, it is always less than 5%, with the largest difference being in the POTRF test case. In the latter case, the slowest heuristic is nearly 10% slower than the fastest.

We provide the average relative differences between heuristics for the Cholesky factorization in Chameleon (POTRF) in Fig. 13. This is the application in which the choice of heuristic has the most impact.

We observe that the heuristics PRWS and PURWS are the ones that give the best execution times, while the NTC (NOD Time Combination) heuristic is the one leading to the worst execution times for this application.

This study suggests that the choice of heuristic typically has an impact of less than 1% on the resulting execution time. The highest impact we measure is less than 10% slowdown between the fastest and the slowest heuristic in the POTRF test case. The impact of the choice of heuristic is, therefore, limited compared to the one of the scheduler. In practice, this implies that application developers can rapidly assess the performance of Heteroprio on their application only by testing one heuristic (typically with a ±1% makespan confidence interval). Additionally, once a user determines that Heteroprio is efficient for their application, they can further fine-tune the scheduler by benchmarking different heuristics and choosing the best one.

## CONCLUSION

Our study presents six heuristics that allow finding efficient priorities automatically. These heuristics rely on the properties of the tasks, such as their makespan or their potential to release other tasks. We show that they can be used to set up the Heteroprio scheduler automatically and achieve high-performance. We perform a theoretical evaluation of the heuristics, which demonstrates that on random graphs the makespan difference between them is typically less than 10%. We then evaluate these heuristics on four real applications. In a first study, we show that AutoHeteroprio usually offers performance comparable to that of Heteroprio, the main difference being that AutoHeteroprio is fully automatic while Heteroprio needs expert-defined priorities. In a second study, we show that the choice of heuristic has a limited impact on the execution time in these four applications (±1% execution time). This suggests that real executions are generally more

impacted by the choice of scheduler (Heteroprio, HEFT, etc.) than by the choice of heuristic within the AutoHeteroprio framework. These two studies support the contributions that AutoHeteroprio brings to HPC developers in practice. On the one hand, it can be quickly tested since it is fully automatic, contrary to Heteroprio. On the other hand, it can be further tuned if needed, either by choosing among the 28 existing heuristics or by designing a new heuristic by hand.

The study for the Chameleon GEMM (Fig. 7) emphasizes that AutoHeteroprio can be slower than Heteroprio. This is due to the overhead induced by the cost of fetching the data of the tasks and of computing the priorities. It can theoretically be removed by enabling the automatic mode in the first runs and inputting the resulting priorities in the normal semi-automatic Heteroprio mode. Hence, the following runs would be as fast as possible, as long as the automatic priorities are efficient. We plan on adding a feature that would automatically switch Heteroprio to a non-automatic mode once enough data are fetched. This feature should take the performance AutoHeteroprio to the same level as Heteroprio in the eyes of application developers who use AutoHeteroprio as a fully automatic scheduler.

The most problematic cases are the ones where neither Heteroprio nor Auto-Heteroprio succeed in achieving high-performance, e.g., in PaStiX (Fig. 10) and Chameleon factorizations (Fig. 9). These cases suggest that our approach could be more general. We are working on a new scheduling paradigm where the tasks are not grouped by type anymore. In this paradigm, the same heuristics as in AutoHeteroprio are used, but for each task individually, rather than for the whole bucket. This induces problems, as the total number of tasks can become very large, while the number of buckets is assumed to be limited.

In summary, this study shows that the semi-automatic scheduling paradigm of Heteroprio can be extended to a fully automatic paradigm where user interaction is no more required for guiding the scheduler. It leads to the creation of heuristics that have been tested and validated in an execution simulator. These heuristics allow the conception of AutoHeteroprio: an automatic version of Heteroprio in StarPU. Our benchmarks show that the AutoHeteroprio alternative usually performs as well as (and sometimes better than) its semi-automatic counterpart.

## APPENDIX

### Heteroprio execution example

To understand the theoretical principle of Heteroprio, let us consider the example DAG shown in Fig. 14 and the associated costs of Table 5.

There are three task types (A, B, and C). We assume that within a type, all tasks have the same costs. Let us consider a case where there are 2 CPU workers and 1 GPU worker. For the sake of simplicity, let us assume that the tasks are selected in a predefined order: CPU-1 pops a task if there is one available, then GPU-1, and then CPU-2. In practice, this order is not known in advance. In a real StarPU execution, there is a "prefetch" mechanism that ensures that a worker can start the job immediately after the dependencies are satisfied.

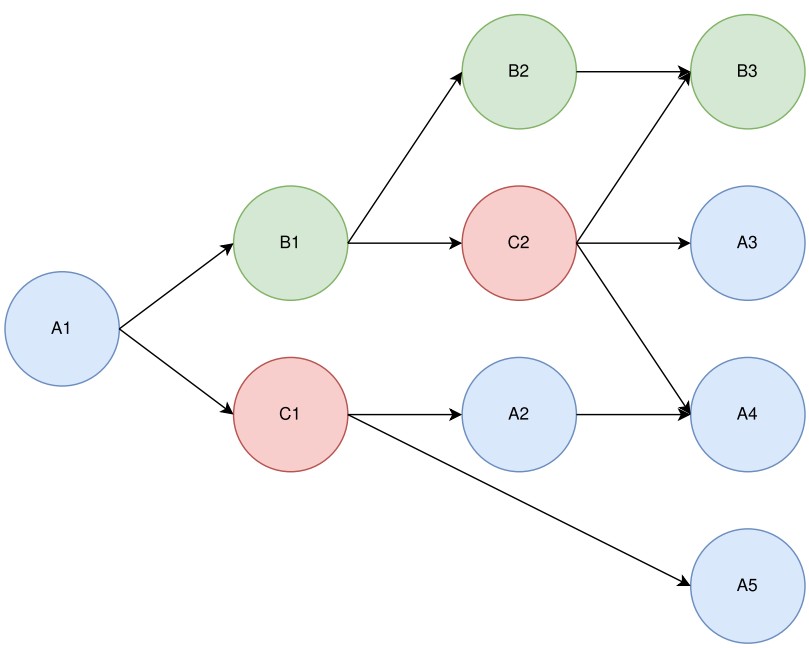

**Figure 14 Example of a DAG with three task types (blue, red, and green).**

**Table 5 Example execution times of for the two processing unit types (CPU/GPU) and the three task types (1, 2, and 3).**

| Architecture Task | CPU (s) | GPU (s) |
|---|---|---|
| A | 1 | 2 |
| B | 2 | 1 |
| C | 1 | 1 |

**Table 6 Example priorities for a configuration of two processing unit types (CPU/GPU) and three task types (A, B, and C).**

| Architecture Task | CPU | GPU |
|---|---|---|
| 1 | B-C-A | A-C-B |
| 2 | A-C-B | B-C-A |
| 3 | C-A-B | B-C-A |

Intuitively, "A" tasks seem to be better suited for CPUs, which execute them twice as fast, whereas "B" tasks seem better suited for GPUs. The "C" tasks do not, seem to have particular affinities. In our model, whatever priorities we set, A1 is always executed by CPU-1. Also, C2 seemingly has great importance, since it has three successors.

Let us test what happens under three different priority lists. In Table 6, we show three different test cases.

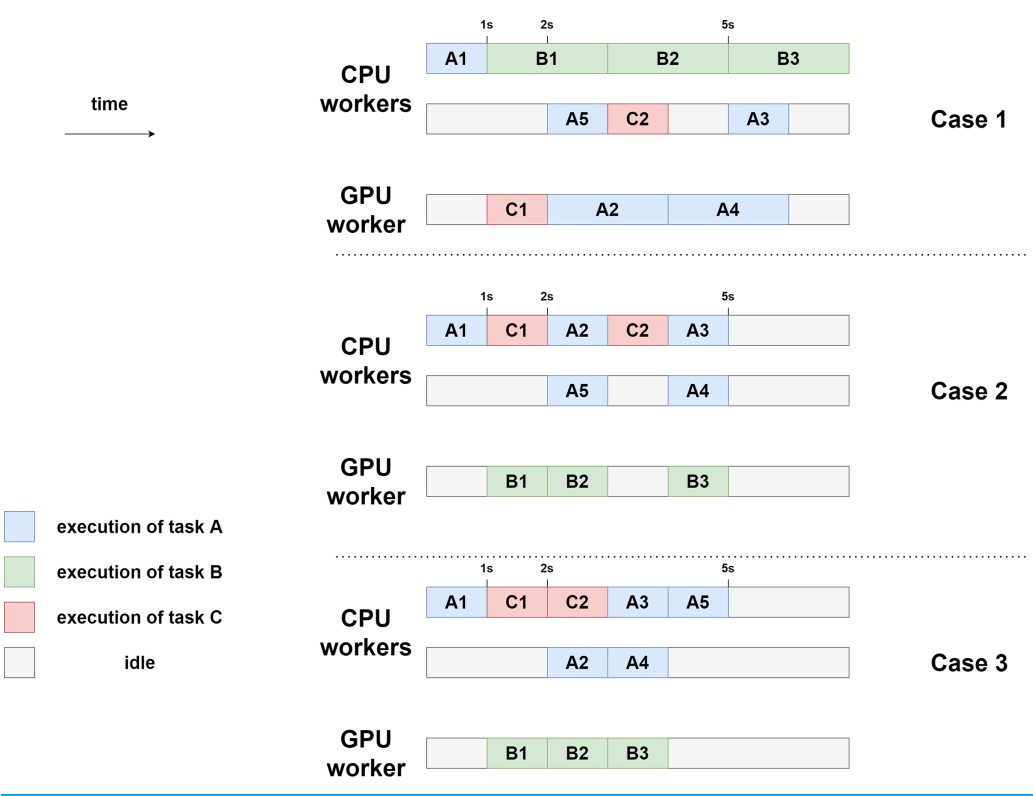

**Figure 15 Example executions for the three different priority settings using two CPU workers and one GPU worker.**

**Table 7 Tested priorities and slowdown factors for the POTRF operation (Chameleon's Cholesky factorization).**

| Name | CPU priorities | Cuda priorities | Slowdown factors | | |
|---|---|---|---|---|---|
| | | | trsm | syrk | gemm |
| base | portf – splgsy – trsm – syrk – gemm | trsm – syrk – gemm | 11.0 | 26.0 | 29.0 |
| inverted-CPU | trsm – syrk – gemm – portf – splgsy | trsm – syrk – gemm | 11.0 | 26.0 | 29.0 |
| inverted-GPU | portf – splgsy – gemm – syrk – trsm | gemm – syrk – trsm | 11.0 | 26.0 | 29.0 |
| **low-factors** | **portf – splgsy – trsm – syrk – gemm** | **trsm – syrk – gemm** | **2.0** | **2.0** | **4.0** |
| high-factors | portf – splgsy – trsm – syrk – gemm | trsm – syrk – gemm | 25.0 | 45.0 | 49.0 |

**Note:**
Row in bold are the priorities used in our benchmarks.

In case 1, B is the highest-priority task type on CPUs and A is the lowest one. On GPUs, the priorities are reversed. For both processors, C is the median priority. In this first case, the slowest architectures are intentionally promoted. For case 2 and case 3, we promote the fastest architectures. The difference between the two is that the priorities in case 2 are mirrored compared to the ones in case 1, whereas in case 3, we exchange the C and A types in the CPU. The idea of this swap is to favor the execution of C2 which has numerous successors.

The three executions are schematized in Fig. 15. In this model, the makespan of case 1 is lower (7 s) than the one of case 2 and case 3 (5 s). Here, case 2 and case 3 are equivalent in

**Table 8 Tested priorities and slowdown factors for the GEMM operation (Chameleon's matrix/matrix multiplication).**

| Name | CPU priorities | Cuda priorities | Slowdown factors gemm |
|---|---|---|---|
| base | plrnt – gemm | gemm | 29.0 |
| inverted | gemm – plrnt | gemm | 29.0 |
| **low-factors** | **plrnt – gemm** | **gemm** | **1.0** |
| high-factors | plrnt – gemm | gemm | 40.0 |

**Note:**
Row in bold are the priorities used in our benchmarks.

**Table 9 Tested priorities and slowdown factors for the GEQRF operation (Chameleon's QR factorization).**

| Name | CPU priorities | Cuda priorities | Slowdown factors ormqr | tpmqrt |
|---|---|---|---|---|
| **base** | **geqrt – tpqrt – plrnt – lacpy – laset – ormqr – tmpqrt** | **ormqr – tmpqrt** | **10.0** | **10.0** |
| inverted_CPU | ormqr – tmpqrt – geqrt – tpqrt – plrnt – lacpy – laset | ormqr – tmpqrt | 10.0 | 10.0 |
| inverted_others | lacpy – laset – geqrt – tpqrt – plrnt – ormqr – tmpqrt | ormqr – tmpqrt | 10.0 | 10.0 |
| low-factors | geqrt – tpqrt – plrnt – lacpy – laset – ormqr – tmpqrt | ormqr – tmpqrt | 2.0 | 2.0 |
| high-factors | geqrt – tpqrt – plrnt – lacpy – laset – ormqr – tmpqrt | ormqr – tmpqrt | 22.0 | 22.0 |

**Note:**
Row in bold are the priorities used in our benchmarks.

**Table 10 Tested priorities and slowdown factors for the PaStiX.**

| Name | CPU Priorities | Slowdown factors cblk_gemm | blok_trsm | blok_gemm |
|---|---|---|---|---|
| base | olve_blok_{trsm – gemm} – cblk_{getrf1d – gemm} – blok_{getrf – trsm – gemm} | 1.0 | 10.0 | 10.0 |
| **better_factors** | **solve_blok_{trsm – gemm} – cblk_{getrf1d – gemm} – blok_{getrf – trsm – gemm}** | **4.0** | **2.0** | **3.0** |
| inverted_groups | blok_{getrf – trsm – gemm} – cblk_{getrf1d – gemm} – solve_blok_{trsm – gemm} | 1.0 | 10.0 | 10.0 |
| better_factors_higher | solve_blok_{trsm – gemm} – cblk_{getrf1d – gemm} – blok_{getrf – trsm – gemm} | 5.0 | 3.0 | 4.0 |
| low-factors | blok_{getrf – trsm – gemm} – cblk_{getrf1d – gemm} – solve_blok_{trsm – gemm} | 1.0 | 1.0 | 1.5 |
| high-factors | blok_{getrf – trsm – gemm} – cblk_{getrf1d – gemm} – solve_blok_{trsm – gemm} | 5.0 | 15.0 | 5.0 |
| Cuda priorities | base cblk_gemm – blok_{trsm – gemm} - - - **better_factors cblk_gemm – blok_{trsm – gemm}** | – | – | – |
| inverted_groups | blok_{trsm – gemm} – cblk_gemm | – | – | – |
| better_factors_higher | cblk_gemm – blok_{trsm – gemm} | – | – | – |
| low-factors | blok_{trsm – gemm} – cblk_gemm | – | – | – |
| high-factors | blok_{trsm – gemm} – cblk_gemm | – | – | – |

**Note:**
Row in bold are the priorities used in our benchmarks.

terms of makespan. In case 3, however, CPU-2 and GPU-1 are freed sooner (after 4 s of execution) than in case 2, where they are still working after 5 s of execution. Case 3 can, therefore, be seen as potentially better. This emphasizes the difficulty of finding heuristics automatically. Indeed, some tasks should be prioritized depending on their execution time, but others should be prioritized because they have particular importance in the execution graph (as C in our example).

## Manual priority settings

For the results we provide in "Evaluation on real applications", the non-automatic Heteroprio executions use manual priorities. These priorities are selected from a careful benchmark for each application. We follow different strategies for choosing them and we provide the different priorities that we test: Table 7 for POTRF, Table 8 for GEMM, Table 9 for GEQRF, and Table 10 for PaStiX. For QrMumps and Scalfmm, we use the already existing priorities set in the code.

## ACKNOWLEDGEMENTS

Experiments presented in this report were carried out using the PlaFRIM experimental testbed, supported by Inria, CNRS (LABRI and IMB), Université de Bordeaux, Bordeaux INP and Conseil Régional d'Aquitaine (see https://www.plafrim.fr/).

### Funding

This work was supported by the ICPS Team from the ICube laboratory, the CAMUS team from Inria Nancy, and by the Department of Mathematics and Computer Science, University of Strasbourg. The funders had no role in study design, data collection and analysis, decision to publish, or preparation of the manuscript.

### Grant Disclosures

The following grant information was disclosed by the authors:
ICPS Team from the ICube laboratory.
CAMUS Team from Inria Nancy.
Department of Mathematics and Computer Science, University of Strasbourg.

### Competing Interests

The authors declare that they have no competing interests.

### Author Contributions

- Clément Flint conceived and designed the experiments, performed the experiments, analyzed the data, performed the computation work, prepared figures and/or tables, authored or reviewed drafts of the article, and approved the final draft.
- Ludovic Paillat performed the experiments, analyzed the data, performed the computation work, prepared figures and/or tables, and approved the final draft.
- Bérenger Bramas analyzed the data, authored or reviewed drafts of the article, and approved the final draft.

### Data Availability

The code is available at GitLab: https://gitlab.inria.fr/cflint/auto_heteroprio_analysis.

## Supplemental Information

Supplemental information for this article can be found online at http://dx.doi.org/10.7717/peerj-cs.969#supplemental-information.

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
