# Peer review of "Automated prioritizing heuristics for parallel task graph scheduling in heterogeneous computing"

_PeerJ Computer Science, doi:10.7717/peerj-cs.969_

## Round 0.1 · original submission · Major Revisions

Please carefully address all the concerns raised by the reviewers and re-submit the revised version in the due time. Thank you.

·

Basic reporting

Please let me to comment some guidelines about this research work.

This research work is based on to already published works (I mean these references Bramas, B. (2016). Bramas, B. (2019). Bramas, B., Flint, C., and Paillat, L. (2021). Bramas, B., Helluy, P., Mendoza, L., and Weber, B. (2020).
), although the authors do not express it explicitly; And it looks for gaps for improvement as the authors comment: "the previous work used static priorities".

This research work do not try to obtain priorities for each task but for each type of task.

This work uses a set of metrics to measure the performance of the system proposed.

And now please let me to comment some suggestions:

When you work on parallel architectures, you use specific programming languages and methodologies for these types of architectures. Could you explicitly indicate which programming languages or methodologies you used on the hardware architectures? In the "Software" section I only found the applications that were parallelized.
The programming language is important because it helps us achieve the desired results. And future works may propose other programming languages.


Although the makespan metric is used as a metric (the most important) to measure the performance of the proposed system, I didn´t find a formal formulation of it in 3.1 Relevant metrics section. I comment that makespan is defined on page 6 as basic definition only. A formal definition can help us understand how you get the results of the makespan metric in section 4.2 Evaluations.

Experimental design

I consider that the evaluations are developed correctly; they are extensive and interesting. As Figures 2 and 3 show, LAAutoheteroprio is superior in acceleration and overcome in both experiments.

Validity of the findings

I want to highlight the findings in the experiments. As you explain in section 4.2.3.

StarPU overhead is relatively high as it has been designed to handle tremendous data (what we already know).
The use of scheduler is only relevant when the expected gained time is greather than the scheduler's overhead.

My comments: it is clearly observable that the results are biased by the architecture and the structure of the application task.
Although extensive experimentation was carried out (I am referring to figures 5-12) there is a high dependency on the hardware configuration (as you express on before section 4.2.3), so I recommend describing each phrase widely, of the contributions of this study in the section: "Introduction”, in order to better understand the contributions of carrying out this research work.

Additional comments

I was trying access this reference Bramas, B., Flint, C., and Paillat, L. (2021). auto-heteroprio analysis. https://gitlab.inria.fr/cflint/auto_heteroprio_analysis.
(On Click):
https://gitlab.inria.%20fr/cflint/auto_heteroprio_analysis
But I couldn’t access. It is important reference because we can view the code of the form how you can create the graph.


Regarding contribution: "We describe different heuristics to obtain priorities;" at the final of the paper (in the conclusions section), you wrote: "Our results also demonstrate that our automatic strategy is typically competitive with manual priorities".
Then I do not find differences between automatic and manual priorities (according to your comments and experimentation), so you do not seek to obtain priorities but to match these?
Can you explain this justification further?

Do you have any future jobs? Any work after this investigation?
Generally, a research projects a set of works after the work presented.
Could you comment in a section of future works (but not mandatory) some possible investigations to be carried out?

I find discrepancy in the spelling of the planner name LaHeteroprio and LAHeteroprio. I was looking if you were indicating another planner, but it is the same planner; I strongly suggest correcting this name.

Reviewer 2 ·

Basic reporting

The paper is generally well written, deals with an interesting topic and provides a useful contribution to the state of the art.
However, the following issues need to be addressed:
1. As the main part, the heuristics and the corresponding formulas should be described in more detail. Also in Sections 3.1 and 3.2, the metrics and heuristics contain the term v_i which is not introduced.
2. A more detailed discussion of the results would be useful. Especially in the Sections 4.2.3 and 4.2.4, more explanations may help to clarify the improvements of your approach. A legend should be added to Figures 5-12.

Experimental design

There are some important details missing in the manuscript:
1. Which heuristic is used to determine the automatic priorities in the evaluation in Section 4.2?
2. In Section 3.3, it is stated that the task execution times are taken from previous task executions of the same type. Is there a strategy to select the values of the first task execution or are they chosen randomly?

Validity of the findings

1. The authors should point out the impact of their work.
What is the advantage of the automatic strategy compared to manual priorities?
Does it reduce the effort for the user or does it lead to better schedules (which is the case for some applications and for some not)?
2. There are some contradictions in the manuscript:
- In line 353, it is stated that the choice of heuristic is expected to have a significant impact, whereas the Conclusion contains the opposite sentence.
- In the lines 426 and 427, it is stated that "using automatic priorities does not hurt performance", although they are slower in some cases and also for some following results, e.g. the solve step of PaStiX.

Additional comments

Minor issue:
- Table 4 contains some misplaced characters

---

## Round 0.2 · accepted · Accept

Congratulations, your manuscript has been recommended for publication.

·

Basic reporting

no comment

Experimental design

no comment

Validity of the findings

no comment

Additional comments

This was the second revision of the article. All suggestions and corrections proposed in my review were taken care of. I consider that the review and suggestions to the article were completed.

Reviewer 2 ·

Basic reporting

The revised version of the paper addresses the issues that I raised in my previous review of the original submission.
In particular, the results are evaluated and explained much more detailed in the revised version.
Together with the rewritten Section 2.2., this makes the paper much easier to read and the reader gets a better understanding of the scheduling problem, the heuristics and the approach.

Experimental design

The missing information about the heuristic used in the evaluation and strategies for estimating the execution time for the first run of a task have been added in the revised version.

Validity of the findings

With the added paragraphs, the impact and novelty of the work are described much better.
The revised version points out the main improvements and drawbacks of the approach which makes the contribution of the paper much clearer.

---

## Author Rebuttal · Round 0.2

# Automated prioritizing heuristics for parallel task graph scheduling in heterogeneous computing.
## Answer to Reviewers

Clément Flint, Ludovic Paillat, Bérenger Bramas

March 6, 2022

## 1   Summary

We thank both reviewers for their valuable comments. We have edited our manuscript in accordance with their remarks. We also provide a *latex-diff* version of the paper that emphasizes the modifications.

## 2   Reviewer 1 (Apolinar Velarde Martínez)

> *(1) Please let me to comment some guidelines about this research work.*
> *This research work is based on to already published works (I mean these references Bramas, B. (2016). Bramas, B. (2019). Bramas, B., Flint, C., and Paillat, L. (2021). Bramas, B., Helluy, P., Mendoza, L., and Weber, B. (2020). ), although the authors do not express it explicitly; And it looks for gaps for improvement as the authors comment: "the previous work used static priorities".*
> *This research work do not try to obtain priorities for each task but for each type of task.*

Thank you for providing this summary. All these statements are correct.

> *(2) This work uses a set of metrics to measure the performance of the system proposed.*

Indeed, we measure the performance with the makespan or the speedup/slowdown.

> *(3) And now please let me to comment some suggestions:*
> *When you work on parallel architectures, you use specific programming languages and methodologies for these types of architectures. Could you explicitly indicate which programming languages or methodologies you used on the hardware architectures? In the "Software" section I only found the applications that were parallelized. The programming language is important because it helps us achieve the desired results. And future works may propose other programming languages.*

Thank you for the suggestion, let me clarify the framework. All this work lies within the framework of the task-based parallelization method. That is to say that an application is (manually) divided into tasks that are sent to the runtime system which will handle the coherent parallel execution. As our work focuses on the scheduler within the runtime system (StarPU for this study), we mostly emphasize the methodologies that relate to Heteroprio and the automatic finding of priorities (as the methodologies/languages used by the applications or StarPU are out of our control). This point was tackled in the sections "1) Introduction", "2.1.2) Heteroprio overview" and "2.2) Formalization". To address your remark, we add details in the "Software" paragraph which make it clear that the applications rely on StarPU and the task-based method.

> *(4) Although the makespan metric is used as a metric (the most important) to measure the performance of the proposed system, I didn´t find a formal formulation of it in 3.1 Relevant metrics section. I comment that makespan is defined on page 6 as basic definition only. A formal definition can help us understand how you get the results of the makespan metric in section 4.2 Evaluations.*

Absolutely, this was an omission. The section "2.2) Formalization" has been rewritten. We divide it into two subsections: the first "2.2.1) General scheduling problem" that relates to general scheduling metrics and the second "2.2.2) Heteroprio automatic configuration problem" that shows the specificities of the scheduling in Heteroprio. We provide a proper definition of the makespan in "2.2.1) General scheduling

problem".

> *(5) Experimental design I consider that the evaluations are developed correctly; they are extensive and interesting. As Figures 2 and 3 show, LAAutoheteroprio is superior in acceleration and overcome in both experiments. Validity of the findings I want to highlight the findings in the experiments. As you explain in section 4.2.3.*
> *StarPU overhead is relatively high as it has been designed to handle tremendous data (what we already know). The use of scheduler is only relevant when the expected gained time is greather than the scheduler's overhead.*
> *My comments: it is clearly observable that the results are biased by the architecture and the structure of the application task. Although extensive experimentation was carried out (I am referring to figures 5-12) there is a high dependency on the hardware configuration (as you express on before section 4.2.3), so I recommend describing each phrase widely, of the contributions of this study in the section: "Introduction", in order to better understand the contributions of carrying out this research work.*

We agree that the first version is not clear enough concerning the contribution of this study. We add a paragraph (line 58 in the diff) in the introduction to emphasize that the contribution lies within the automation of Heteroprio and the preservation of high-performance executions. We also add summaries at the end of "4.2.3) Comparison with other schedulers" and "4.2.4) Comparison of different heuristics in AutoHeteroprio" that aim at underlining the practical consequences of these results for the typical application developer. Finally, the conclusion has been rewritten and now summarizes the most important aspect of this work.

> *(6) Additional comments I was trying access this reference Bramas, B., Flint, C., and Paillat, L. (2021). auto-heteroprio analysis. https://gitlab.inria.fr/cflint/auto_heteroprio_analysis. (On Click): https://gitlab.inria.%20fr/cflint/auto_heteroprio_analysis But I couldn't access. It is important reference because we can view the code of the form how you can create the graph.*

We are not able to reproduce this. When we click on the link in the pdf file, we arrive at the correct address.

> *(7) Regarding contribution: "We describe different heuristics to obtain priorities;" at the final of the paper (in the conclusions section), you wrote: "Our results also demonstrate that our automatic strategy is typically competitive with manual priorities". Then I do not find differences between automatic and manual priorities (according to your comments and experimentation), so you do not seek to obtain priorities but to match these? Can you explain this justification further?*

Our work aims at generating efficient priorities automatically. We do not particularly want the automatically generated priorities to match the static ones, as we do not know with certainty which priorities are the best. Also, as the automatic priorities change throughout the execution, it is hard to compare them against the static ones. The main contribution of this work lies within the full automatization of Heteroprio. The added paragraph in the introduction should clarify this point.

> *(8) Do you have any future jobs? Any work after this investigation? Generally, a research projects a set of works after the work presented. Could you comment in a section of future works (but not mandatory) some possible investigations to be carried out?*

The future work is now described in more detail in the conclusion.

> *(9) I find discrepancy in the spelling of the planner name LaHeteroprio and LAHeteroprio. I was looking if you were indicating another planner, but it is the same planner; I strongly suggest correcting this name.*

The references to LaHeteroprio are now consistent.

# 3 Reviewer 2 (Anonymous)

*(10) The paper is generally well written, deals with an interesting topic and provides a useful contribution to the state of the art. However, the following issues need to be addressed: 1. As the main part, the heuristics and the corresponding formulas should be described in more detail. Also in Sections 3.1 and 3.2, the metrics and heuristics contain the term $v\_i$ which is not introduced.*

Thank you for these encouraging comments! The term $v\_i$ is now defined in 2.2.1. Concerning the heuristics and their corresponding formulae, we provide additional details that should help the reader understand the sense of the formulae.

*(11) 2. A more detailed discussion of the results would be useful. Especially in the Sections 4.2.3 and 4.2.4, more explanations may help to clarify the improvements of your approach. A legend should be added to Figures 5-12.*

The added details at the end of 4.2.3 and 4.2.4 should summarize the implications of these two studies. As for the legends, we do not see what information could be added directly to the diagrams, as all the values are explicitly mentioned (the makespan and its unit, the name of the configuration, and the name of the schedulers). We have clarified the captions of the corresponding figures to specify that the boxes represent the distribution of the obtained execution times.

*(12) Experimental design There are some important details missing in the manuscript: 1. Which heuristic is used to determine the automatic priorities in the evaluation in Section 4.2?*

Yes, the information was missing. We now specify the choices for the two studies:

- in section 4.2.2 (lines 497-498 in the diff)

- in section 4.2.3 (lines 560-564 in the diff)

*(13) 2. In Section 3.3, it is stated that the task execution times are taken from previous task executions of the same type. Is there a strategy to select the values of the first task execution or are they chosen randomly?*

Yes, there are strategies for estimating execution times on the first execution of a task. We have designed two strategies that are now described in lines 366-370. The default strategy is used for our benchmarks, although its impact should be negated by the fact that the first runs are not taken into account with Heteroprio.

*(14) Validity of the findings 1. The authors should point out the impact of their work. What is the advantage of the automatic strategy compared to manual priorities? Does it reduce the effort for the user or does it lead to better schedules (which is the case for some applications and for some not)?*

Thank you for pointing out this issue. The added paragraph in the introduction and the conclusion are now more explicit about the impact of this work. The first impact is indeed that the programming effort is reduced. The second impact is that it can sometimes be faster than state-of-the-art schedulers, which indicates that this approach can perform well in real executions.

*(15) 2. There are some contradictions in the manuscript: - In line 353, it is stated that the choice of heuristic is expected to have a significant impact, whereas the Conclusion contains the opposite sentence.*

Yes, the evaluation based on emulated executions suggests that the heuristic might have an impact of $\pm$ 10% on the execution time, but the evaluation on real applications contradicts this hypothesis and suggests that the choice of heuristic is less impactful in real-time executions than in simulated ones.

*(16) - In the lines 426 and 427, it is stated that "using automatic priorities does not hurt performance", although they are slower in some cases and also for some following results, e.g. the solve step of PaStiX.*

Yes, but there are also cases where automatic priorities are faster. That is why we claim that "Generally, it does not harm performance". The main goal here is to discuss the potential overhead that additional treatments can induce (here, the computation of the priorities). We rephrased the sentence: "It appears that using automatic priorities does not always harm performance".

*(17) Additional comments Minor issue: - Table 4 contains some misplaced characters*

Corrected.

# 4  Miscellaneous

We provide the following other changes:

- we put every table in a separate tex file;

- we improve the English;

- we change the BibTeX file so that the reference "Optimization and parallelization of the boundary element method for the wave equation in time domain" can now spread on multiple lines.